# Dissecting features of epigenetic variants underlying cardiometabolic risk using full-resolution epigenome profiling in regulatory elements

Fiona Allum [1,2], Åsa K. Hedman[3], Xiaojian Shao[1,2], Warren A. Cheung[1,2,12], Jinchu Vijay [1,2], Frédéric Guénard[4], Tony Kwan [1,2], Marie-Michelle Simon[1,2], Bing Ge[1,2], Cristiano Moura[5], Elodie Boulier[1,2], Lars Rönnblom[6], Sasha Bernatsky[5], Mark Lathrop[1,2], Mark I. McCarthy [7,8,9], Panos Deloukas [10], André Tchernof[11], Tomi Pastinen[1,2,12], Marie-Claude Vohl[4] & Elin Grundberg[1,2,12]

Sparse profiling of CpG methylation in blood by microarrays has identified epigenetic links to common diseases. Here we apply methylC-capture sequencing (MCC-Seq) in a clinical population of ~200 adipose tissue and matched blood samples ($N_{total}$~400), providing high-resolution methylation profiling (>1.3 M CpGs) at regulatory elements. We link methylation to cardiometabolic risk through associations to circulating plasma lipid levels and identify lipid-associated CpGs with unique localization patterns in regulatory elements. We show distinct features of tissue-specific versus tissue-independent lipid-linked regulatory regions by contrasting with parallel assessments in ~800 independent adipose tissue and blood samples from the general population. We follow-up on adipose-specific regulatory regions under (1) genetic and (2) epigenetic (environmental) regulation via integrational studies. Overall, the comprehensive sequencing of regulatory element methylomes reveals a rich landscape of functional variants linked genetically as well as epigenetically to plasma lipid traits.

[1] Department of Human Genetics, McGill University, Montréal, QC H3A 0C7, Canada. [2] McGill University and Genome Quebec Innovation Centre, Montréal, QC H3A 0G1, Canada. [3] Department of Medicine Solna, Cardiovascular Medicine Unit, Karolinska Institute, Stockholm 171 76, Sweden. [4] Institute of Nutrition and Functional Foods (INAF), Université Laval, Québec, QC G1V 0A6, Canada. [5] Department of Epidemiology, McGill University, Montréal, QC H3A 1A2, Canada. [6] Department of Medical Sciences, Uppsala University, Uppsala 751 85, Sweden. [7] Oxford Centre for Diabetes, Endocrinology and Metabolism, Churchill Hospital, University of Oxford, Old Road, Headington, Oxford OX3 7LJ, UK. [8] Wellcome Centre for Human Genetics, University of Oxford, Roosevelt Drive, Oxford OX3 7BN, UK. [9] Oxford NIHR Biomedical Research Centre, Oxford University Hospitals NHS Foundation Trust, John Radcliffe Hospital, Oxford OX3 9DU, UK. [10] William Harvey Research Institute, Barts and The London School of Medicine and Dentistry, Queen Mary University of London, Charterhouse Square, London EC1M 6BQ, UK. [11] Québec Heart and Lung Institute, Université Laval, Québec, QC G1V 0A6, Canada. [12] Present address: Children's Mercy Hospitals and Clinics, Kansas City, MO 64108, USA. Correspondence and requests for materials should be addressed to E.G. (email: egrundberg@cmh.edu)

Complex diseases such as obesity and type 2 diabetes (T2D) are caused by joint action of predisposing genetic and environmental factors[1–4]. Heritability measures of obesity-related traits such as BMI have shown that the genetic contribution is likely only ~30–40%[5]—pointing towards a larger impact than previously estimated by environmental effects.

CpG methylation has been shown to be disrupted in disease states[6,7] and by environmental modifiers[8,9]. As such, assessment of CpG methylation changes through epigenome-wide association studies (EWAS) enables us to connect environment and genetics[10,11] to phenotype and disease[12]. Circulating lipid profiles are clinically applied in cardiometabolic risk assessment[4], providing indications of metabolic complications among healthy and obese individuals[13]. Although past EWAS efforts have successfully identified lipid-associated loci with roles in metabolic processes[14–18], we have shown the importance of using disease-targeted tissues for functional interpretation of disease loci due to the preferential mapping of identified variants to tissue-specific regulatory elements[11,19]. This is an important observation considering that most EWAS to-date have studied whole-blood tissue using targeted arrays (e.g., Illumina 450 K array), which under-represent distal regulatory regions (e.g., enhancers) and bias towards promoter regions. In fact, promoters are largely uninformative in EWAS due to the invariable state of resident CpGs across individuals[11], partly due to insufficient sensitivity measures in DNA methylation assessments.

To overcome this limitation, we implemented the methylC-capture sequencing (MCC-seq) approach permitting simultaneous methylome and genotype profiling in regulatory regions at high resolution[20]. A pilot adipose tissue EWAS of triglyceride (TG) levels identified novel TG-linked methylation variation within enhancers. MCC-Seq was also applied across various tissues in hundreds of donors and demonstrated stronger enrichment of GWAS SNPs underlying allele-specific methylation within disease-linked tissues—emphasizing the importance of utilizing appropriate tissues to decipher not only epigenetic variants but genetic variants[21].

Here, we present a large next-generation sequencing (NGS)-based EWAS applying MCC-Seq on adipose tissue and blood samples derived from a clinically relevant cohort of obese individuals. We link ~1.3 M dynamic CpGs to blood plasma lipids and map positional trends of lipid-linked CpGs within functional elements. We highlight the ability of MCC-Seq to fine-map EWAS signals through replication in the large MuTHER adipose cohort and apply integrative approaches to identify disease-associated epigenetic variants linked to regulatory effects, further providing insight into metabolic disease etiology. We further show features of the metabolic-disease-linked methylome by assessing the contribution of genetic factors and use these tabulated associations to fine-map cardiometabolic-risk-associated GWAS SNPs.

## Results

**Adipose tissue epigenetic variants linked to plasma lipids.** CpG methylation was profiled in visceral adipose tissue (VAT) from 199 severely obese individuals (BMI > 40 kg m$^{-2}$; 60% female) undergoing bariatric surgery (IUCPQ, Université Laval; Supplementary Table 1; see Methods section). We applied the MCC-Seq protocol querying up to 3.3 M CpGs mapping to adipose tissue regulatory regions[20] (see Methods section). We focus on a conservative set of highly covered (33×) and variable sites corresponding to 1.3 M CpGs (see Methods section) that exhibited mainly (55%) hypomethylated states (<20% average methylation) with a smaller proportion (10%) being hypermethylated (>80% average methylation).

We associated CpG methylation at the 1.3 M sites in adipose tissue with circulating plasma lipid levels, i.e., triglycerides (TG), HDL-cholesterol (C), LDL-C, and total cholesterol (TC) (see Methods section), applying a generalized linear model accounting for sequencing depth, age and BMI. Controlling for bias and inflation of our test-statistics was achieved using the Bayesian method BACON[22], noting an improvement in the inflation factor (lambda) after correction across all trait-associations (Supplementary Figures 1-4). In total, methylation levels at 1230 (FDR 10%; corrected $p < 3.52 \times 10^{-5}$) and 615 (FDR 5%; corrected $p < 9.25 \times 10^{-6}$) CpGs were associated to at least one lipid trait (Supplementary Figure 5). We subsequently refer to "lipid-CpGs" as those reaching significant lipid associations at FDR 10% (Supplementary Data 1). Overall, 13% of lipid-CpGs were linked to more than one lipid trait (Supplementary Figure 5). By assessing the inter-individual variability of lipid-CpGs, these sites also depicted a more variable state than the full set of 1.3 M CpGs tested (Supplementary Figure 6).

**Positioning of lipid-CpGs within regulatory elements.** Identified lipid-CpGs were annotated using adipose tissue hypo-methylated footprints—low-methylated regions (LMRs) and unmethylated regions (UMRs)[20,23]—as indicators of regulatory elements. We previously characterized these methylated footprints[23], showing co-localization of adipose tissue LMRs and UMRs with the H3K4me1 active enhancer and H3K4me3 active promoter marks, respectively, from primary human adipocytes (NIH Roadmap Consortium). In all subsequent analyses, we refer to LMRs and UMRs as putative enhancers and promoters, respectively. We additionally characterized these adipose tissue regulatory regions in terms of their genomic lengths and discovery CpG densities, where we noted putative enhancers were shorter and less densely populated than promoters (Supplementary Table 2). Mimicking our previous findings[20], lipid-CpGs were enriched in putative adipose enhancers (26% of lipid-CpGs versus 17% in background; Fisher's exact test throughout; Fisher's $p = 6.6 \times 10^{-13}$) while being less likely to map to putative promoters (40% of lipid-CpGs versus 54% in background; Fisher's $p < 2.2 \times 10^{-16}$; Supplementary Figure 7). The set of lipid-CpGs was then restricted to include only those mapping to adipose tissue regulatory regions not shared with other tissues (i.e., whole-blood; see Methods section) and showed stronger enrichment patterns at enhancers (13% of lipid-CpGs versus 7% in background; Fisher's $p = 9.9 \times 10^{-13}$). Additionally, we noted a reversal of trends as lipid-CpGs were enriched in adipose-specific promoters (10% of lipid-CpGs versus 6% in background; Fisher's $p = 8.1 \times 10^{-11}$; Supplementary Figure 7). Of note, these localization patterns appear to be independent of CpG methylation variability at interrogated sites (Fisher's $p < 1.1 \times 10^{-7}$; top 25th percentile; Supplementary Figure 7). In total, we identified 264 putative adipose enhancers (LMRs) and 303 promoters (UMRs) harboring lipid-CpGs, of which 341 are shared elements and 226 are adipose-specific elements. These 567 regulatory elements were carried forward for further analyses (Supplementary Data 1; Fig. 1).

Given the high-density coverage of CpG methylation obtained through MCC-Seq, we investigated differences in positional trends of lipid-CpGs within adipose tissue hypomethylated footprints (see Methods section). Focusing first on all discovery CpGs mapping to the 264 LMRs, lipid-CpGs located more towards the mid-point of putative enhancers compared to all CpGs (Fig. 2a). CpGs locating to UMRs (within +/−1.5 Kb of a transcription start site (TSS); 139/303 UMRs) exhibited a bimodal distribution flanking the TSS similar to the background with a slight peak shift downstream of the TSS further into the gene

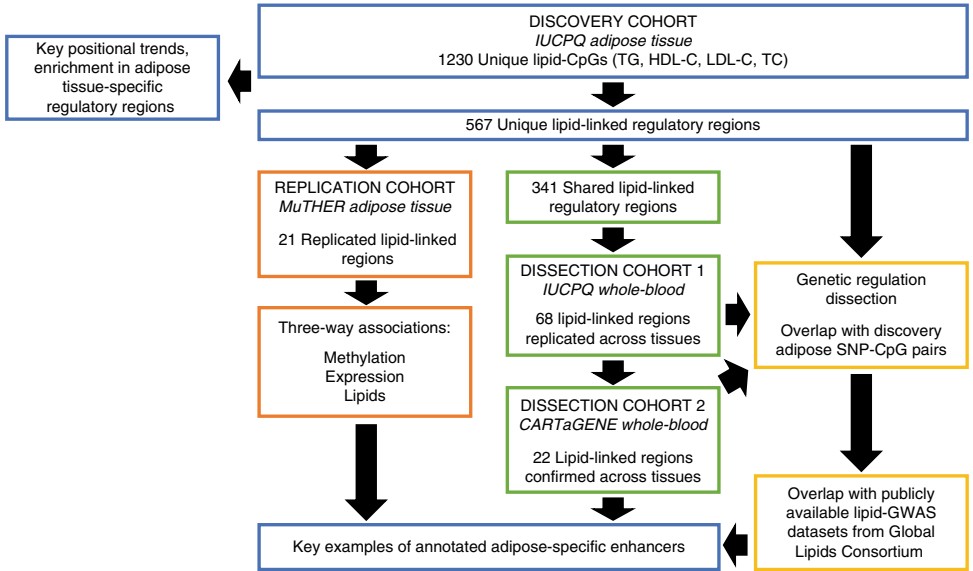

**Fig. 1** Study flow chart. Overview of included study cohorts and follow-up analyses to characterize identified lipid-linked adipose tissue regulatory regions

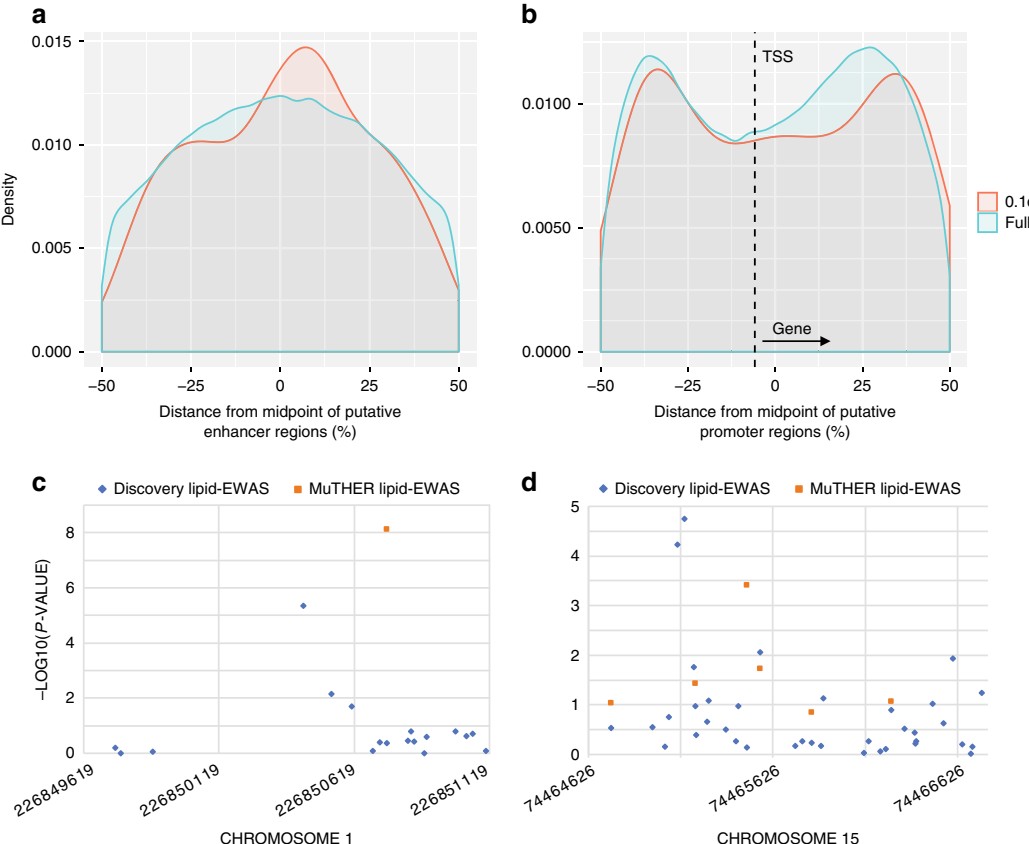

**Fig. 2** Positional mapping of lipid-CpGs within adipose tissue regulatory elements. Specific positional trends of significant lipid-CpGs (FDR 10%) merged across all studied lipids traits (i.e., triglycerides, HDL-C, LDL-C, and total cholesterol) were investigated at adipose regulatory regions. Positions of CpGs were tabulated as the percent distance from the midpoint of elements (genomic distance from midpoint (bp)/length of element(bp)*100) and collapsed to summarize positional trends over all assessed elements. Positional trends are shown for **a** CpGs mapping to LMRs ($N = 225,771$) and **b** CpGs mapping to UMRs within $+/-1.5$ kb of transcription start sites (TSS) not depicting bivalent gene transcription orientations (taking gene orientation into account; $N = 418,246$). The fine-mapping potential of MCC-Seq over array-based methods is exemplified in **c** a replicated HDL-linked enhancer region (chr1:226849619-226851122) and **d** a replicated LDL-linked promoter region (chr15:74464626-74466792), where we noted top discovery lipid-CpGs to mimic trends noted in **a** and **b** at adjacent sites to signals identified from the large-scale 450K-based MuTHER study

body (Fig. 2b). To rule out potential technical biases explaining these observations, we assessed the mean coverage of reads within these elements and found that although lipid-CpGs have higher coverage than all assessed CpGs, the coverage does not differ based on the position of lipid-CpGs within the elements per se (Supplementary Figure 8).

Next, we contrasted the capacity of MCC-Seq to capture lipid-CpGs within adipose tissue regulatory regions over alternative methods such as the Illumina 450 K[14-18] and EPIC arrays. As a whole, the EPIC and 450 K arrays captured only 17 and 6% of the total percent CpGs profiled in LMRs by MCC-Seq and 29 and 19% of those mapping to UMRs, respectively. These percentages dropped further when focusing on CpGs typed on the array-based methods directly overlapping MCC-Seq CpGs (Supplementary Table 3). Positional trends of CpGs in both arrays showed a depletion of coverage within putative promoters downstream of the TSS (Supplementary Figure 9)—regions towards the gene body where we showed lipid-CpGs to be enriched (Fig. 2b).

**Replication of lipid-linked adipose regulatory regions**. We then validated the 567 adipose regulatory regions mapping with lipid-CpGs in the MuTHER cohort ($N \sim 650$ individuals) where sub-cutaneous adipose tissue CpG methylation levels were profiled on the 450 K array[11] and associated to the same lipid traits under investigation (TG, HDL-C, LDL-C, and TC)[18]. Of the 567 high-lighted regulatory regions, only 365 (64%) were covered by the 450 K array. In line with design biases of the 450 K array, a higher proportion of adipose tissue promoter regions (269/303 UMRs; 89%) than enhancer regions (96/264 LMRs; 36%) contained at least one 450K array CpG. Using Bonferroni cutoff (taking into account each trait individually and with same direction of effect), we found the highest replication rate for TG-UMRs where 17% (13/76) of the regions were also associated with TG in the validation cohort. All replicated regions ($N = 21$) are presented in Supplementary Data 2.

To assess the potential of MCC-Seq to fine-map EWAS signals, we focused on the 16 of 21 replicated regulatory regions containing at least 2 discovery lipid-CpGs where one of these overlapped the top MuTHER lipid-CpG. Here, 15/16 (94%) elements harbored stronger lipid associations at discovery CpGs that didn't directly overlap the top MuTHER lipid-CpG positions (Supplementary Data 2). We then investigated the localization of the "fine-mapped" discovery lipid-CpGs compared to their nearby MuTHER lipid-CpGs within the adipose tissue regulatory elements. All the "fine-mapping" discovery CpGs located at the mid-point of adipose tissue LMRs ($+/-20\%$ from mid-point), representing a slight increase in proportion over their paired MuTHER CpGs (2/3 CpGs). This pattern is similar to the observed positional mapping trends for the full set of lipid-CpGs at LMRs, which exhibited a mid-point shift compared to all CpGs assessed (Fig. 2a, c). Likewise, "fine-mapping" discovery CpGs mapping to adipose UMRs showed that these CpGs tended to locate in greater numbers (7/12; 58%) than their paired MuTHER CpGs (5/12 CpGs; 42%) within the bimodal positional peaks ($+20$ to $+45\%$ or $-20$ to $-45\%$ from mid-point) previously observed for lipid-CpGs at UMRs (Fig. 2b, d). Both of these fine-mapping trends did not reach nominal significance most likely owing to the small number of observations and the additional bimodal pull of the fine-mapping exhibited at the putative promoter regions.

**Functional annotation of lipid-CpGs**. Replicated lipid-linked adipose hypomethylated regulatory regions were characterized by performing transcription factor binding site (TFBS) motif

analyses (see Methods section). Focusing on replicated UMRs harboring lipid-CpGs ($N = 16$ regions) and excluding LMRs due to their small number ($N = 5$), TFBS linked to adipogenesis and/ or obesity related metabolic-complications were enriched, with members of the STAT family[24–26] STAT5A[27], STAT1 and STAT3[28] being most significant, followed by NFIB[29,30] and RUNX1[31,32] (Supplementary Table 4). We further noted that STAT5A, STAT3, and NFIB showed higher levels of expression in adipose tissues over whole-blood in the GTEx Consortium data (GTEx portal; November 2017; Supplementary Figures 10-12) with the strongest evidence for NFIB expression. We confirmed adipocyte-specific expression of NFIB through differential expression analyses of purified human adipocytes from both subcutaneous and visceral depots versus various blood cell types (>14.0-fold change; $p < 3.98 \times 10^{-236}$; see Methods section).

Next, replicated lipid-linked adipose tissue regulatory regions ($N = 5$ LMRs; $N = 16$ UMRs) were functionally annotated by incorporating matching adipose tissue gene expression data from the MuTHER cohort[11] (see Methods section). As many as 16/21 (76%) lipid-associated regions showed significant association between the methylation status of one of their resident CpGs and the expression levels of at least one cis-located gene (FDR 10%; range 1–9 associated genes/region; within $+/-1$Mb; Supplementary Data 3)—representing a 1.9-fold change in effect over all testable regulatory regions (10,141/26,050 regions; Fisher's $p = 0.00104$). All 16 regulatory regions depicting associations to at least one gene also exhibited stronger effects on gene expression at non-adjacent genes—with an average absolute distance of ~522 kb to their most correlated gene compared to ~33 kb to the transcribed region of their most proximal gene. A greater proportion of these replicated lipid-associated regulatory regions (11/16 regions; 69%) correlated to the expression levels of more than one gene compared to the background (4673/10,141 regions; 46%; Fisher's $p = 0.08$).

We assessed whether the genes ($N = 44$) for which expression levels were associated with methylation status at replicated lipid-linked regions ($N = 16$) were also independently linked to the same plasma lipid phenotypes (see Methods section). As many as 77% (30/39) of testable genes linked to 15 replicated lipid-associated regulatory regions showed additional association to the same lipid trait under investigation in the expected direction of effect (Supplementary Data 4).

Restricting to genes listed in the GWAS SNP catalog ($N = 20/$ 30; accessed September 2018), we observed that 6/20 (30%) genes associating to lipid-linked regulatory regions also showed association to metabolic-related phenotypes, revealing an enrichment of obesity-linked traits compared to the full catalog (692/ 15,815 genes; 4%; 6.9-fold change; Fisher's $p = 0.00016$; Supplementary Data 4; see Methods section). Ingenuity pathway analysis (see Methods section) of all 30 highlighted genes showed $G\alpha q$ Signaling as the most significantly associated function within this gene set ($p = 6.94 \times 10^{-5}$; Supplementary Table 5). Interestingly, two of the four genes mapped to this pathway were regulated by the same lipid-regulatory element which we follow-up in more detail below.

**Tissue-specificity of lipid-linked regulatory regions**. To gain insight into the potential tissue-specific nature of epigenetic signatures associated to disease, we interrogated whether lipid-linked signals mapping to regulatory regions are detectable across tissues within a study population by profiling CpG methylation in whole-blood from a matching set of samples ($N = 206$) from the obese IUCPQ cohort (Supplementary Table 1). We linked whole-blood methylation status to the same circulating plasma lipid levels (see Methods section) and successfully typed 565 out of the

567 regulatory regions harboring discovery adipose tissue lipid-CpGs in whole-blood, of which 340 were shared and 225 adipose-specific (i.e., not shared to whole-blood) elements (see Methods section). Globally at the same significance threshold (using Bonferroni cutoff for each trait individually and with same direction of effect), lipid-associations at shared regulatory elements replicated at a significantly higher rate (46/340 replicated lipid-linked regions; 14%) than adipose-specific elements (12/225 replicated regions; 5%; Binomial test $p = 9.0 \times 10^{-9}$). Lipid-associations at shared putative promoters (i.e., UMRs) were more likely to replicate across tissues than at shared enhancer regions—with 35/221 (16%) lipid-linked UMRs compared to 11/119 (9%) LMRs replicating in whole-blood. Specifically, we were able to validate associations at 4/39 (10%) TG-LMRs, 2/39 (5%) HDL-LMRs, 7/34 (21%) LDL-LMRs, 4/26 (15%) TC-LMRs, 10/64 (16%) TG-UMRs, 7/69 (10%) HDL-UMRs, 11/77 (14%) LDL-UMRs, and 14/57 (25%) TC-UMRs in whole-blood (Supplementary Data 5). Previous studies have indicated the importance of accounting for differences in biological outcome of environmental and genetic effects on DNA methylation at the tissue level[18], thus we performed the replication across adipose tissue to whole-blood by also allowing different directions of effect across tissues. Here, we were able to validate additional associations at 1/39 (3%) HDL-LMRs, 1/34 (3%) LDL-LMRs, 6/64 (9%) TG-UMRs, 4/69 (6%) HDL-UMRs, 10/77 (13%) LDL-UMRs, and 3/57 (5%) TC-UMRs in whole-blood (Supplementary Data 5). Taken together, we identified 68 adipose tissue regulatory regions (13 putative enhancers and 55 promoters) showing evidence for tissue-shared lipid-associations.

Pathway analysis of the 52 genes directly overlapping the 68 tissue-independent regulatory regions (Supplementary Table 6) revealed the adipogenesis pathway as the top significantly associated function (IPA $p = 3.1 \times 10^{-3}$; see Methods section). Among the genes highlighted within this pathway, we noted (1) the serine/threonine kinase *AKT1* overlapping a shared promoter region (chr14:105260438-105262714) harboring CpGs positively correlated to both LDL-C and TC levels; (2) the histone deacetylase *HDAC4* mapping with an intergenic enhancer region (chr2:240240338-240241584) containing CpGs depicting negative associations to HDL-C in adipose tissue that were reversed in whole-blood; (3) *BMP4* overlapping a shared promoter region (chr14:54418956-54424030) where CpGs were negatively associated to TG levels. We further highlighted lipid-associated promoter regions at the following cardiometabolic risk-related loci; growth factor *GDF7*, kinase *CERK*, *VGLL3* and ATP-binding cassette transporter *ABCC5*.

Next, we investigated how the lipid-linked and tissue-shared regulatory regions identified in a clinical population associate with the same traits independently of obesity status. CpG methylation was profiled by MCC-Seq in whole-blood from a population-based (N = 137) cohort (CARTaGENE; https://cartagene.qc.ca/; Supplementary Table 1), again linking whole-blood methylation status to circulating plasma lipid levels (Methods). Overall, we found 22/68 (32%) regions to be associated with the same lipid trait under investigation in the population-based cohort (Supplementary Data 6). However, contrasting adipose lipid-associations that replicated in whole-blood with the same (N = 46 regions) versus opposing (N = 28 regions) directions of effects (N = 17/46 regions; 37% vs. N = 5/28 regions; 18%) showed a marked difference in the replication rate (>2-fold change) indicating the possibility of the latter being more specific to the clinical condition.

**Genetic contribution to lipid-CpG methylation variability.** We previously validated the ability and accuracy of MCC-Seq to

| Table 1 Genetic regulation on lipid-linked adipose regulatory regions | |
|---|---|
| **Lipid-linked regulatory regions** | **Genetic regulation enrichment (fold-change)** |
| All lipid-linked elements (N = 567) | 1.5 |
| Adipose-specific elements (N = 226) | 1.1 |
| Tissue-shared elements (N = 341) | 1.7 |
| Tissue-shared elements validated in blood cohort 1 (N = 68) | 2.1 |
| Tissue-shared elements validated in blood cohort 1 and 2 (N = 22) | 2.2 |

provide genotyping information over target regions[20], which we used here to study genetic effects on CpG methylation. Using this inferred genetic dataset, we integrated recently tabulated SNP-CpG associations (metQTL) in *cis* (+/−250 kb) for a subset of the adipose discovery cohort[21]. First, we confirmed our previous findings[11] that SNPs associated with CpG methylation are enriched in the vicinity of their linked CpGs (Supplementary Figure 13). Second, we investigated the level of genetic regulation among lipid-associated regulatory regions and noted a large fraction to be partly under genetic regulation. In line with previous studies[11,18,33], we observed that 64% (362/567) of lipid-associated elements depicted a significant SNP-CpG association (FDR 10%) compared to only 44% (22101/50759) in the background (Fisher's $p < 2.2 \times 10^{-16}$; Table 1). We further found that this enrichment was maintained when accounting for overall methylation variability (top 25% variable CpG methylation status across all individuals; 194/406 lipid-linked regions versus 4763/17593 in background; Fisher's $p < 2.2 \times 10^{-16}$).

We queried whether the identified lipid-linked regulatory regions have different levels of genetic contribution depending on their tissue-specificity and contrasted the elements unique to adipose (N = 226) versus those shared across tissues to whole-blood (N = 341; Table 1). We observed an enrichment in association to *cis*-SNPs only at shared regulatory elements (N = 251/341 regions; 74%; Fisher's $p < 2.2 \times 10^{-16}$; Table 1). Restricting to the subset of 68 lipid-associated shared regulatory regions that were further validated in the matched whole-blood cohort, we noted an increase in observed genetic variation contribution corresponding to as much as 93% (N = 63/68 regions; Fisher's $p < 2.2 \times 10^{-16}$; Supplementary Data 5; Table 1). Finally, we further filtered the list of lipid-linked regulatory regions to only contrast those that in addition to being validated in the matched whole-blood cohort were also significantly associated to lipids in the independent population-based cohort (Supplementary Data 6). Here, we found a striking enrichment with 21/22 (95%) of these tissue-independent and obese-status-independent regions to be under genetic regulation (Fisher's $p = 3.3 \times 10^{-7}$).

To assess whether these genetically controlled lipid-linked epigenetic loci overlap GWAS loci, we incorporated GWAS SNPs for the same four lipid traits under study from the large-scale efforts of the Global Lipids Genetics Consortium[34]. We focused on lead SNPs associated with methylation of CpGs mapping to the 362 lipid-linked regulatory regions. Intersecting these SNPs and/or their proxies ($r^2 > 0.8$) with the fully released dataset of GWAS SNPs at nominal significance, we noted an enrichment at lipid-linked regulatory regions for all lipid traits; TG (3.7-fold; Fisher's $p = 3.4 \times 10^{-16}$), HDL-C (4.4-fold; Fisher's $p < 2.2 \times 10^{-16}$), LDL-C (4.3-fold; Fisher's $p < 2.2 \times 10^{-16}$) and TC (4.1-fold; Fisher's $p < 2.2 \times 10^{-16}$). Enrichment trends were maintained at a more stringent significance cutoff for GWAS SNPs ($p = 5.0 \times 10^{-8}$) albeit with lower statistical confidence due to smaller numbers (Fisher's $p < 0.05$).

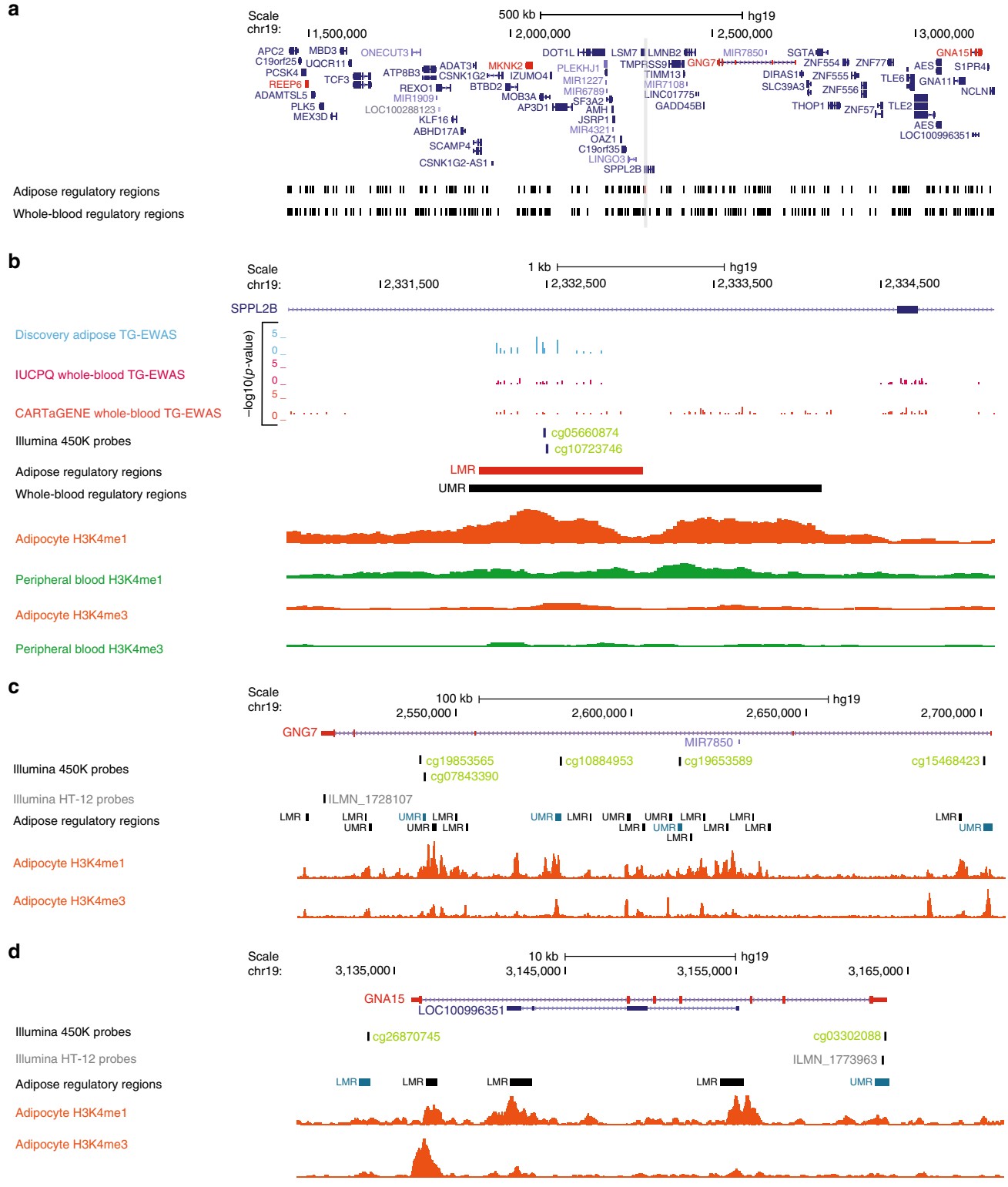

**Regulation of lipid-linked adipose-specific enhancers**. Genetic regulation of lipid-linked regulatory elements is pronounced among regions shared across tissue to whole-blood whereas adipose-specific regions exhibited a larger component of environmentally-driven regulation. Specifically, we found no evidence of genetic associations for 115/226 (51%) lipid-linked regulatory regions active in adipose but not whole-blood. Among

these lipid-linked regulatory regions with non-genetic regulatory effects, we followed-up on an adipose-specific putative enhancer (chr19:2332094-2333076) harboring adipose lipid-CpGs linked to TG levels in our discovery and replication samples. (Supplementary Data 2). This enhancer region maps to the first intragenic region of *SPPL2B*—a locus with no reported associations to cardiometabolic risk (Fig. 3a, b). We initially highlighted the

**Fig. 3** TG-linked adipose-specific regulatory region shows putative pleiotropic effects. A top discovery TG-CpG (chr19:2332436; corrected $p = 2.4 \times 10^{-5}$; sky blue track) replicated by multiple nearby MuTHER TG-CpGs (cg05660874; $p = 5.1 \times 10^{-10}$; cg10723746; $p = 1.0 \times 10^{-8}$; light green track) locates within an adipose-specific enhancer region (chr19:2332094-2333076) overlapping the first intron of *SPPL2B* (LMR; shown in red in **a** the broad and **b** zoomed-in view). Methylation levels at cg05660874 and cg10723746 show associations to cis-locating *REEP6*, *MKNK2*, *GNG7*, and *GNA15* (highlighted in red in **a**), which in turn exhibit associations to TG levels in the MuTHER cohort with *GNG7* and *GNA15* showing the strongest links (**c** *GNG7*; cg05660874 versus ILMN_1709247; $p = 4.9 \times 10^{-5}$; cg10723746 versus ILMN_1709247; $p = 3.1 \times 10^{-8}$; ILMN_1709247 versus TG; $p = 1.2 \times 10^{-12}$; **d** *GNA15*; ILMN_1773963 versus cg10723746; $p = 1.5 \times 10^{-17}$; ILMN_1773963 versus cg05660874; $p = 1.5 \times 10^{-16}$; ILMN_1773963 versus TG; $p = 1.5 \times 10^{-18}$). We show evidence for a co-regulation network between these two genes and the enhancer region by highlighting associations between 450 K array probes (light green tracks in **b**–**d**) locating to several regulatory regions (shown in red in **b** and teal in **c**, **d**) and expression levels of **c** *GNG7* and **d** *GNA15* in MuTHER. We show a lack of whole-blood lipid-EWAS signals at the enhancer of interest (**b**), which is supported by the adipocyte-specific nature of chromatin signatures observed at the locus (Roadmap Epigenomics Consortium; adipocyte nuclei donor 92 shown in orange versus peripheral blood donor TC015 shown in green)

region for harboring TG-linked methylation in the discovery cohort near the mid-point of the enhancer region (chr19:2332436; corrected $p = 2.4 \times 10^{-5}$; Fig. 3b)—mimicking positional trends for lipid-CpGs at this type of element. The positive correlation of methylation to TG levels at this region was validated in the large-population based MuTHER cohort at nearby CpGs (cg05660874; $p = 5.1 \times 10^{-10}$; cg10723746; $p = 1.0 \times 10^{-8}$; Supplementary Data 2; Fig. 3b). Confirming earlier results for the characterization of adipose putative enhancers[23], overlapping the intragenic region with adipocyte-specific H3K4me1 and H3K4me3 (Roadmap; donor 92) showed co-localization of the highlighted adipose-specific LMR with the H3K4me1 enhancer mark (Fig. 3b). This was not observed in peripheral blood (Roadmap; donor TC015) ChIP-Seq data as H3K4me1 peaks were absent, indicating the adipose-specific nature of the regulatory marks (Fig. 3b). This observation corroborates the lack of replication of epigenetic regulation from whole-blood EWAS at this element (Supplementary Data 5; Fig. 3b).

Integrating the MuTHER cohort expression data (see Methods section) revealed a lack of significant epigenetic-association to expression levels of the *SPPL2B* locus. In line with the trend reported above, we instead noted that expression levels of *GNA15*—located 803 kb downstream of the putative enhancer region—were the most correlated (ILMN_1773963 versus cg10723746; $p = 1.5 \times 10^{-17}$; ILMN_1773963 versus cg05660874; $p = 1.5 \times 10^{-16}$; Fig. 3d). We further observed links to expression levels of *GNG7* (179 kb downstream), *REEP6* (834 kb upstream) and *MKNK2* (281 kb upstream; Supplementary Data 3; Fig. 3a). Expression levels of these four genes were also associated with TG levels in the MuTHER cohort, with *GNA15* and *GNG7* exhibiting the strongest relationships (*GNA15*; ILMN_1773963; $p = 1.5 \times 10^{-18}$; *GNG7*; ILMN_1728107; $p = 1.2 \times 10^{-12}$; Supplementary Data 4; Fig. 3c, d)—corroborating the link between regulation of these loci with levels of TG and disease state. Supporting a co-regulation network between these genes is the strong correlation between the 450 K array probes located at several regulatory regions at these genes and the expression products of *GNG7* and *GNA15* interchangeably (Supplementary Data 7; Fig. 3c, d). *GNA15* and *GNG7* both encode for G-protein subunits with suggested roles for *GNA15* in heart failure[35] and glucose homeostasis[36] and for *GNG7* in coronary artery calcification[37] (Supplementary Data 4). Both of these genes mapped to the top IPA disease-linked function of *Gαq Signaling* for genes under regulation by replicated lipid-linked regulatory regions (IPA $p = 6.94 \times 10^{-5}$; Supplementary Table 5). Taken together, this may suggest that the identified adipose-specific regulatory region has pleiotropic effects regulating both *GNA15* and *GNG7* expression, resulting in additive disease risk.

Although we observed a lack of enrichment for genetic associations among lipid-linked regulatory regions active in adipose but not whole-blood, we identified 111 regions under genetic regulation. To exemplify this, we focused on an element mapping to an intragenic region of *GALNT2* (chr1:230312462-230313455) showing both epigenetic and genetic associations to HDL-C (Fig. 4; Supplementary Data 1). Specifically, we showed that this lipid-linked regulatory region (corrected $p = 2.0 \times 10^{-5}$) is under tight genetic regulation with seven CpGs associating to multiple SNPs ($N = 21$) flanking this element (Supplementary Data 8; Fig. 4b). These lead SNPs were in high LD ($r^2 > 0.9$) with an HDL-linked GWAS SNP[34] (Global Lipids Consortium; rs627702; $p = 5.0 \times 10^{-24}$) located 11 kb downstream of the enhancer (Fig. 4). Of note, this HDL-linked GWAS SNP was independent of the top GWAS SNP reported by the Global Lipids Consortium study for this same trait, which locates upstream of the enhancer region (rs4846914; $p = 4.0 \times 10^{-41}$; Fig. 4a)[34]. Genetic effects at this enhancer were supported by conditional analysis where absence of lipid-CpG association was noted when genotypes were included in the model with rs2760537 being the most prominent (corrected $p = 4.3 \times 10^{-2}$; $q = 0.78$; see Methods section). Dissecting results with whole-blood EWAS showed the adipose-specific nature of HDL-association at this region (Fig. 4). This enhancer is also not covered on the 450 K array, representing a novel avenue for HDL-association to epigenetic variants. In addition, we found no evidence of *cis*-eQTLs (GTEx Consortium) linking genetic variants at this locus to gene expression (Fig. 4b). This observation in combination with the lack of a strong adipocyte-specific H3K27ac signature at this enhancer indicates a possible poised or primed region state, supporting efforts highlighting the superior molecular value provided by epigenetics traits over gene expression alone[38]. The glycosyltransferase *GALNT2* locus itself has previously been associated to metabolic syndrome[39], TG levels[40,41] and type 2 diabetes[42], with our current results supporting additional links to cardiometabolic disease through putative epigenetic regulation.

## Discussion

We recently introduced MCC-Seq[20] as a cost-effective and flexible platform for simultaneous DNA methylation and genotype interrogation in large-scale cohorts, permitting targeted and dense profiling of active methylomes within disease-relevant tissues. Here, we apply MCC-Seq in a comprehensive epigenome-wide study of plasma blood lipids (including TG, HDL-C, LDL-C, and TC) and identify 567 lipid-linked regulatory regions in visceral adipose tissue. We combine a stringent statistical correction method (BACON) with a more lenient FDR threshold (10%) and perform detailed follow-ups on regions replicating across adipose tissue depots and across tissue types (whole blood) using the classical Bonferroni approach. This strategy allowed us to present an expanded resource of cardiometabolic risk-linked epigenetic loci.

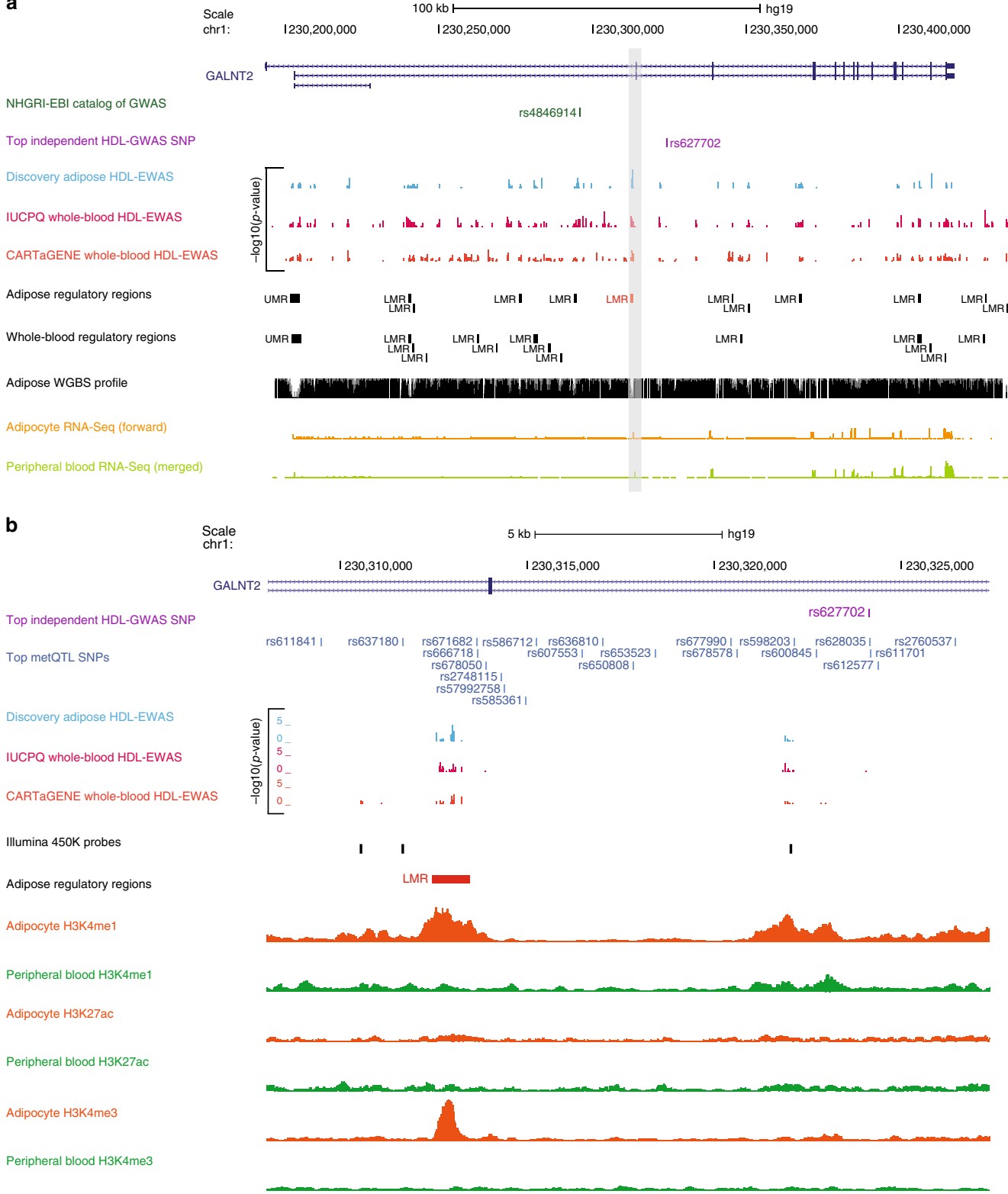

We confirm current epigenomics trends where tissue-specific regulatory regions such as enhancers globally appear more likely to contain trait-linked CpGs compared to promoters - emphasizing the importance of targeting these regions to expand our understanding of complex disease biology. The observed under-representation of lipid-associated epigenetic variants within promoters may be attributable to the more tightly regulated and static nature of these elements, where smaller variations may be biologically impactful but harder to statistically identify. Using MCC-Seq for dense single-base resolution profiling at regulatory elements (7 CpGs/LMR and 37 CpG/UMR, respectively) is advantageous by depicting unique positional trends of lipid-

**Fig. 4** HDL-C linked adipose-specific regulatory region under genetic regulation. A discovery HDL-CpG (chr1:230313001; corrected $p = 2.0 \times 10^{-5}$; sky blue track) maps within an intragenic region of *GALNT2* (chr1:230312462-230313455) overlapping an adipose-specific putative enhancer region (LMR; shown in red in **a** the broad and **b** zoomed-in view). The adipose-specific nature of the epigenetic signature at this locus is supported by patterns in adipocyte nuclei (Roadmap Epigenomics Consortium; donor 92 for H3K4me1 and H3K4me3; donor 7 for H3K27ac; orange tracks) versus peripheral blood (Roadmap Epigenomics Consortium; donor TC015; green tracks) chromatin marks as well as from intersecting whole-blood EWAS signals (pink and dark orange tracks). We show that the enhancer region is under extensive genetic regulation by nearby cis-SNPs (gray blue tracks in **b**) that are in high LD ($r^2 > 0.9$) with an HDL-linked GWAS SNP (Global Lipids Consortium; rs627702; $p = 5.0 \times 10^{-24}$; purple tracks), which is independent of the previously reported top HDL-linked SNP at this locus (rs4846914; $p = 4.0 \times 10^{-41}$; dark green track in **a**). We depict a lack of coverage of the 450 K array at this region. Adipocyte-specific (in-house data; light orange track) and peripheral blood RNA-Seq (Roadmap Epigenomics Consortium; donor TC014; light green track) data at the locus is also depicted in **a**

associated epigenetic variants. Key differences are observed in positioning at putative enhancers in contrast to promoters: lipid-associated epigenetic variants show clear enrichment at the mid-point of enhancers whereas they depict a bimodal distribution flanking the TSS of promoters. These observations may reflect the TFBS landscape within regulatory regions with preferential binding of TFs at midpoints or edges dependent on elements. Comparisons of these full-resolution positional trends with those captured by array-based approaches exemplified the limitations of the latter methods to assess CpGs within regulatory regions both in terms of the number of CpGs covered and ascertainment biases due to probe design.

We further demonstrate that NGS-based high resolution CpG profiling in epigenome-wide studies allows for fine-mapping of trait-linked epigenetic signals from large-scale array-based studies. Due to limitations in visceral adipose tissue cohort availabilities, the large MuTHER subcutaneous adipose tissue cohort was used as the best alternative proxy of available data for replication studies, therefore we focused on epigenetic variants stable across these two adipose tissue depots. We highlight a high-confidence set of 21 adipose-specific regulatory regions associated with plasma lipid levels. Identified signals for >90% of lipid-associated regulatory regions were refined, with "fine-mapping" discovery CpGs mimicking positional trends highlighted at adipose regulatory elements. We hypothesize that differences in study design both in terms of adipose tissue depots (visceral versus subcutaneous) and cohort selection (obese versus population-based) between the discovery study and MuTHER cohorts, respectively, may contribute to the observed replication rate. Nevertheless, our TFBS analysis provided insight into potential underlying signaling pathways. Specifically, binding motifs for *NFIB* were enriched at adipose tissue promoter regions mapping with lipid-associated epigenetic variants. Interestingly, *NFIB* has previously been reported to function in glucose transport[29] and also serves as an important regulator of proper adipocyte differentiation as exemplified by preferential mapping to adipocyte-specific or preadipocyte-specific open chromatin peaks[29,30].

While MCC-Seq represents added fine-mapping value for full-resolution methylome assessment, past profiling efforts within the MuTHER cohort have provided us with rich array-based datasets. Linking methylation, expression, and phenotype profiles across ~600 adipose tissue samples, we identify lipid-linked replicated adipose tissue regulatory regions associating to plasma lipid traits and expression levels at unique loci that associate to the same lipid traits. We highlight several obesity-related GWAS loci—*CSK*[43], *SLCO3A1*[44] and *GNG7*[37] and *GNA15*[35,36]—and report several novel genes including *LCN2, ECHS1, IDH2*, and *CD7*. These genes also map to metabolic disease-linked pathways such as the highlighted *Gαq Signaling* known to have a role in adipogenesis through its action in regulating intracellular calcium levels and downstream expression of the master regulators PPARγ and C/EBPα[45,46].

CpG methylation is seen as a proxy linking genetics and environment to disease and phenotype. To further contribute to our understanding of genetic and non-genetic factors impacting complex diseases, we dissect lipid-linked regulatory regions through adipose SNP-CpG associations[21] within the same cohort. As previously observed[11,18,33], a large fraction of lipid-associated regulatory elements is under genetic regulation. These genetic effects are strengthened when restricting to tissue-independent and lipid-linked regions replicating to whole-blood within the same disease-cohort (93%) as well as across cohorts (95%)—hinting at a coordinated mechanistic regulation over these regions across tissues. We highlight an adipose tissue specific putative enhancer on chromosome 1—locating within the first intron of the obesity-linked *GALNT2*[39–42], where methylation levels at this regulatory region are under genetic control by variants within an HDL-linked GWAS locus[34]. Adipocyte-specific histone marks at the locus suggest that the HDL-linked regulatory region represents an adipose-specific poised enhancer and may explain why genetic regulation of this disease locus has not been identified by large eQTL efforts such as from the GTEx Consortium. This finding highlights the importance of studying epigenetic marks such as DNA methylation over gene expression alone.

Building on a previous study[11], we also present an expanded methylation-expression association analysis, permitting us to assess pleiotropic effects of adipose tissue regulatory regions showing association to cardiometabolic risk factors. In line with current chromatin conformational studies, we report that methylation status at lipid-linked regulatory regions shows stronger associations to expression levels of genes locating ~500 kb away, on average. A majority (~70%) of these lipid-linked regulatory elements exhibit putative pleiotropic effects—indicating the occurrence of regulatory networks linked to the disease state. We focus on an adipose tissue-specific TG-linked enhancer region on chromosome 19 showing strong putative effects on the expression levels of two *Gαq Signaling* genes—glucose homeostasis-linked *GNA15*[36] and coronary artery disease-linked *GNG7*[37] located >200 kb upstream of the element. We support observed TG-associations at this region through our three-way associations of methylation, expression and lipids within the MuTHER cohort. We also present evidence for a co-regulation network between regulatory regions mapping to *GNA15* and *GNG7* and the adipose enhancer of interest. Taken together, this may suggest that the identified adipose-specific regulatory region has pleiotropic effects regulating expression of both *GNA15* and *GNG7* resulting in additive disease risk.

In conclusion, our study demonstrates the advantage of NGS-based methylome profiling in disease-relevant tissues to identify complex trait-linked epigenetic variants at high resolution. We show that targeted sequencing approaches enables us to refine methylome landscape features and to further disentangle the genetic versus environmental contributions to complex traits. Our study represents an expanded dataset of cardiometabolic-risk-

linked epigenetic regulatory regions in the disease-relevant adipose tissue. Our findings confirm that integrating cellular phenotypes with disease traits across tissues enables the identification of functional epigenetic variants in regulatory regions linked to complex disease traits.

## Methods

**Sample collections**. We obtained 199 visceral adipose tissue (VAT) samples (males $N = 79$; females $N = 120$) from the Quebec Heart and Lung Institute for our discovery cohort (IUCPQ; Université Laval, Quebec City, Canada). Samples were collected between June 2000 and July 2012 for 1906 severely obese (BMI $> 40$ kg m$^{-2}$) men ($N = 597$) and women ($N = 1309$) undergoing biliopancreatic diversion with duodenal switch[47] at this Institute as previously described[48]. Briefly, subjects fasted overnight before the surgical procedure. Anesthesia was induced by a short-acting barbiturate and maintained by fentanyl and a mixture of oxygen and nitrous oxide. VAT samples were obtained within 30 min of the beginning of the surgery from the greater omentum[48].

We additionally obtained 206 whole-blood samples from the same IUCPQ cohort described above for dissection of adipose epigenetic variants. Blood was collected before surgery.

The sample collection was approved by the Université Laval and McGill University (IRB FWA00004545) ethics committee and performed in accordance with the principles of the Declaration of Helsinki. Tissue banking and the severely obese cohort were approved by the research ethics committees of the Quebec Heart and Lung Institute. All participants provided written informed consent before enrollment in the study.

We included 137 whole-blood samples from the CARTaGENE cohort (https://cartagene.qc.ca/) in the study design for dissection of adipose epigenetic variants. As a whole, the CARTaGENE cohort numbers ~20,000 general population subjects drawn from the province of Québec, Canada. Using bio-banked serum from a random subset ($N = 3600$) of the CARTaGENE cohort, ACPA (anti-citrullinated protein antibody) positive subjects ($N = 69$; 18 with high titers ≥ 60 units, the others with medium titers = 20–59 units) were identified by an enzyme-linked immunosorbent assay (Quanta Lyte, CCP3 IgG: Inova Diagnostics Inc., San Diego, CA). Age and sex-matched ACPA negative subjects ($N = 68$) were randomly selected. ACPA status was not considered as a covariate in this study.

The methylation studies of the samples from CARTaGENE were approved by the McGill University institutional review board, IRB number A04-M46-12B. All participants provided written informed consent before enrollment in the study.

BMI was calculated as weight in kilograms divided by height in meters squared. Plasma total cholesterol (TC), triglyceride (TG), and high-density lipoprotein cholesterol (HDL-C) levels were measured using enzymatic assays. HDL-C was measured in the supernatant following precipitation of very low-density lipoproteins and low-density lipoproteins with dextran sulfate and magnesium chloride. Plasma low-density lipoprotein cholesterol (LDL-C) levels were estimated with the Friedewald formula. Summary of the characteristics are tabulated in Supplementary Table 1.

**MCC-Seq methylation profiling**. Genomic DNA was extracted from the blood buffy coat using the GenElute Blood Genomic DNA kit (Sigma, St. Louis, MO, USA) and quantified using both NanoDrop Spectrophotometer (Thermo Scientific) and PicoGreen DNA methods. The samples were profiled through targeted methylation sequencing as previously described[20]. Briefly, in MCC-Seq a whole-genome sequencing library is prepared and bisulfite converted, amplified and a capture enriching for targeted bisulfite-converted DNA fragments is carried out. The captured DNA is further amplified and sequenced. More specifically, whole-genome sequencing libraries were generated from 700 to 1000 ng of genomic DNA spiked with 0.1% (w/w) unmethylated λ DNA (Promega) previously fragmented to 300–400 bp peak sizes using the Covaris focused-ultrasonicator E210. Fragment size was controlled on a Bioanalyzer DNA 1000 Chip (Agilent) and the KAPA High Throughput Library Preparation Kit (KAPA Biosystems) was applied. End repair of the generated dsDNA with 3′-overhangs or 5′-overhangs, adenylation of 3′-ends, adapter ligation and clean-up steps were carried out as per KAPA Biosystems' recommendations. The cleaned-up ligation product was then analysed on a Bioanalyzer High Sensitivity DNA Chip (Agilent) and quantified by PicoGreen (Life Technologies). Samples were then bisulfite converted using the Epitect Fast DNA Bisulfite Kit (Qiagen), according to the manufacturer's protocol. Bisulfite-converted DNA was quantified using OliGreen (Life Technologies) and, based on quantity, amplified by 9–12 cycles of PCR using the Kapa Hifi Uracil + DNA polymerase (KAPA Biosystems), according to the manufacturer's protocol. The amplified libraries were purified using Ampure Beads and validated on Bioanalyzer High Sensitivity DNA Chips, and quantified by PicoGreen. SeqCap Epi Enrichment System protocols (Roche NimbleGen) were then carried out for the capture step using the previously presented adipose-specific custom panels[20] MetV1 ($N = 113$ discovery adipose samples), MetV2 ($N = 92$ discovery adipose samples; $N = 206$ whole-blood IUCPQ cohort samples) as well as a whole-blood-specific custom panel[49] ($N = 137$ CARTaGENE cohort samples). The hybridization procedure of the amplified bisulfite-converted library was performed as described by the manufacturer, using 1 µg of total input of library, which was evenly divided by the

libraries to be multiplexed, and incubated at 47 °C for 72 h. Washing and recovering of the captured library, as well as PCR amplification and final purification, were carried out as recommended by the manufacturer. Quality, concentration and size distribution of the captured library was determined by Bioanalyzer High Sensitivity DNA Chips. Captures were sequenced on the Illumina HiSeq2000/2500 system using 100-bp paired-end sequencing.

Reads were aligned to the bisulfite converted reference genome using BWA v.0.6.1[50]. We removed (i) clonal reads, (ii) reads with low mapping quality score (<20), (iii) reads with more than 2% mismatch to converted reference over the alignment length, (iv) reads mapping on the forward and reverse strand of the bisulfite converted genome, (v) read pairs not mapped at the expected distance based on library insert size, and (vi) read pairs that mapped in the wrong direction as described by Johnson et al.[51] To avoid potential biases in downstream analyses, we applied our benchmark filtering criteria as follows; ≥5 total reads, no overlap with SNPs (dbSNP 137), ≤20% methylation difference between strands, no off-target reads and no overlap with DAC Blacklisted Regions (DBRs) or Duke Excluded Regions (DERs) generated by the ENCODE project: (http://hgwdev.cse.ucsc.edu/cgi-bin/hgFileUi?db=hg19&g=wgEncodeMapability).

Methylation values at each site were calculated as total (forward and reverse) non-converted C-reads over total (forward and reverse) reads. CpGs were counted once per location combining both strands together. We restricted the analyses to CpGs covered in at least 100 individuals for the IUCPQ cohorts and 50 individuals for the CARTaGENE cohort (due to the smaller cohort size) with more than 10% of these having methylation status above zero and below 100%.

**Epigenome-wide association of plasma lipid levels**. We tested associations between methylation levels of CpGs detected by MCC-Seq with circulating lipid levels (TG, HDL-C, LDL-C, and TC) from the corresponding cohorts using a generalized linear model (GLM) function implemented in R3.1.1. Outliers in lipid levels were identified by setting a cutoff of mean ± 3 * SD and removed from further analysis. Lipid levels not depicting a normal distribution were converted to the log scale (adipose IUCPQ: TG; whole-blood IUCPQ: TG and HDL-C; CARTaGENE: TG, HDL-C, and LDL-C). The response variable (methylation levels) was fitted to a binomial distribution weighted for sequence read coverage at each site and adjusted (1) for age, sex, MCC-Seq panel batch effect and BMI for discovery cohort adipose samples, and (2) for age, sex, blood cell proportions and BMI for the whole-blood IUCPQ and CARTaGENE cohort samples. We remove bias and inflation by applying the bacon correction[22] on the test statistics using default parameters. False-discovery rate (FDR) was calculated with the R/Bioconductor q-value package[52] for each trait individually in the adipose IUCPQ cohort. We set the significance level at FDR 10%. Bonferroni cutoff was used as a significance threshold for dissection with the whole-blood cohorts for each trait individually.

Subcutaneous adipose tissue methylation data from a population-based cohort of 648 female individuals in the TwinsUK/MuTHER cohort was obtained for replication. The samples were profiled on the Illumina 450 K array and normalized as described previously[11]. Associations between 450 K array methylation data ($N = 355,296$ CpG probes) and the four circulating lipid levels under investigation were previously assessed[18] using a linear mixed model taking into account familial relationship, twin zygosity and other cofactors into account (i.e., age, beadchip, BS conversion efficiency, BS-treated DNA input and BMI—expect when assessing BMI itself). Bonferroni cutoff was used as a significance threshold for validation for each trait individually.

**Positional mapping analyses**. We defined un-methylated (UMR) and low-methylated regions (LMR) by mining through whole-genome bisulfite sequencing datasets from adipose and whole-blood samples from the same cohort, separately, as described previously[20,23]. Through these efforts, we reported 20,195 UMRs and 45,065 LMRs for adipose tissue and 19,871 UMRs and 46,159 LMRs for whole-blood samples[20,23]. Adipose-specific regions were previously defined by intersecting adipose and whole-blood hypomethylated regions, where 2342 and 24,687 adipose-specific UMRs and LMRs were tabulated, respectively[20].

Positional trends of CpGs within adipose regulatory elements were assessed restricting to LMRs containing at least 1 CpG ($N = 31,964$) and UMRs containing at least 1 CpG and within $+/-1.5$ kb of transcription start sites (TSS) as well as not depicting bivalent gene transcription orientations ($N = 10,924$). Position of CpGs were tabulated as the percent distance from the midpoint of elements (genomic distance from midpoint (bp)/length of element(bp)*100) and collapsed to make density plots using ggplot2[53] to summarize positional trends over all assessed elements. Gene orientation was additionally taken into account for CpGs mapping to UMRs where UMRs were positioned upstream of genes.

**Transcription factor binding site motif analysis**. Transcription factor binding site (TFBS) motif analysis was performed using the Homer software[54] for lipid-linked UMRs ($N = 16$ regions) replicated in the MuTHER cohort where we excluded replicated lipid-linked LMRs due to their small number ($N = 5$ regions). Default settings were selected with the "given" size option. UMRs harboring replicated lipid-associated CpGs were contrasted against the remaining promoter regions containing interrogated CpGs that lacked nominal significance in the

discovery EWAS for any of the four lipid traits ($N = 912$ UMRs). A Bonferroni $q < 0.05$ cutoff was applied for significance.

**Differential expression analyses**. Peripheral blood mononuclear cells were purified from buffy coats originating from 450 ml blood of healthy blood donors (Uppsala Blood Transfusion Center, Uppsala University Hospital, Sweden), using Ficoll-Paque (GE Healthcare) density-gradient centrifugation. B cells, T cells and monocytes were isolated from dedicated batches of peripheral blood mononuclear cells, using positive selection with CD19+, CD3+, and CD14+ beads (Miltenyi Biotec), respectively, according to the manufacturer's instructions.

RNA isolations were performed using miRNeasy Mini Kit (Qiagen). RNA library preparations were carried out on 500 ng of RNA with RNA integrity number (RIN) > 7 isolated from adipocyte cells extracted from AT[55,56] and blood cells (CD19+, CD3+, and CD14+) using the Illumina TruSeq Stranded Total RNA Sample preparation kit, according to manufacturer's protocol. Final libraries were analysed on a Bioanalyzer and sequenced on the Illumina HiSeq2000 (pair-ended 100 bp sequences). Raw reads were trimmed for quality (phred33 ≥ 30) and length ($N \geq 32$), and Illumina adapters were clipped off using Trimmomatic v. 0.32[57]. Filtered reads were aligned to the hg19 human reference using STAR v.2.5.1b[58]. Raw read counts of UCSC genes were obtained using htseq-count v.0.6.1 (http://www-huber.embl.de/users/anders/HTSeq).

Differential expression analysis was done using DESeq2 v.1.18.1[59] on RNA-seq data from adipocytes isolated from adipose tissue (subcutaneous and visceral) of 20 obese individuals undergoing bariatric surgery (IUCPQ) and different blood cell types ($N = 11$ B cells; $N = 20$ T cells; $N = 20$ monocytes) of healthy European individuals (Uppsala Blood Transfusion Center, Uppsala University Hospital, Sweden). We used stringent cutoffs to define adipocyte-specific expression—requiring log2-fold-change > 2 and $p < 0.05$ across all six comparisons of adipocytes to blood cell types.

**Linking gene expression to methylation in MuTHER cohort**. We expanded on a previously published methylation-expression association analysis performed within the MuTHER cohort[11] to assess possible long-range interactions (+/−1 Mb) for CpGs mapping to both LMRs and UMRs. We restricted to 145,913 450 K CpGs residing in 27,258 adipose regulatory regions and tested for association to 20,326 expression probes (IlluminaHT12) for 602 individuals with matched samples. We used a similar linear mixed-effects model as described previously[11], implemented with the lme4 package[60] lmer() function fitted by maximum likelihood. As before, the linear mixed-effects model was adjusted for both fixed effects (age, beadchip, BS conversion efficiency, BS-treated DNA input) and random effects (family relationship and zygosity) but here we added BMI as an additional covariate. We used a likelihood ratio test to assess the significance of the gene expression effect. The $p$-value of the gene expression effect in each model was calculated from the Chi-square distribution with 1 degree of freedom (df) and $-2 \log$(likelihood ratio) as the test statistic. In total, we tested 4,245,804 methylation to gene expression associations and assessed the false-discovery rate (FDR 10%) using the R/Bioconductor $q$-value package[52].

**Association of gene expression to lipids in MuTHER cohort**. Associations between gene expression levels (IlluminaHT12) and lipid status within the MuTHER cohort were modeled using a linear mixed effects model as described previously[61]. Briefly, the lmer function in the lme4 package[60], was fitted by maximum-likelihood. The linear mixed effects model was adjusted for age and experimental batch (fixed effects) and family relationship (twin-pairing) and zygosity (random effects). A likelihood ratio test was used to assess the significance of the phenotype effect. The $p$-value of the phenotype effect in each model was calculated from the Chi-square distribution with 1 degree of freedom using $-2 \log$(likelihood ratio) as the test statistic.

**Gene enrichment pathway analyses**. Core expression analyses were performed using default settings in the Ingenuity Pathway Analysis software. Only the top 5 canonical pathways are reported. We ran the software on (1) 30 genes showing associations to both replicated adipose lipid-CpGs mapping with adipose regulatory regions and to the same lipid trait independently (see "Functional annotation of lipid-CpGs"), and (2) 52 genes directly overlapping the 68 lipid-linked adipose regulatory regions validated in whole-blood (see "Tissue-specificity of lipid-linked regulatory regions").

**Conditional modeling of HDL-EWAS on SNPs**. Genotypes for the 21 SNPs in the region of interest within *GALNT2* (Supplementary Data 8) were generated for 148 adipose IUCPQ samples. For 56/113 discovery samples profiled via MCC-Seq using MetV1, genotypes were typed on the high-density genotyping using the Illumina HumanOmni2.5–8 (Omni2.5) BeadChip according to protocols recommended by Illumina. For 92 discovery samples profiled via MCC-Seq using MetV2, genotypes were inferred using the Bis-SNP software[62], a bisulfite-sequencing variant caller, with default parameters: "-T BisulfiteGenotyper -stand_call_conf 20 -stand_e-mit_conf 0 -mmq 30 -mbq 17 -minConv 0" and with dbSNP 137 as prior SNP information. The aligned bam files were used as input file and the hg19 was used as the reference genome.

Conditional modeling of HDL-association at chr1:230313001 for the 21 SNPs in the region of interest within *GALNT2* (Supplementary Data 8) was carried out independently for each SNP by adding the genotype status as a covariate in the GLMs as described in "Epigenome-wide association of plasma lipid levels" above.

**Reporting summary**. Further information on experimental design is available in the Nature Research Reporting Summary linked to this article.

## Data availability

The methylation and expression data from the MuTHER cohort have been deposited in the ArrayExpress, https://www.ebi.ac.uk/arrayexpress/ (accession no. "E-MTAB-1866" and "E-TABM-1140". Lipid-EWAS results from the adipose and whole-blood IUCPQ cohorts as well as the whole-blood CARTaGENE cohort can be visualized in the UCSC Genome Browser by adding the following URL to "My Hubs": https://emc.genome.mcgill.ca/myHub/hub_adipose.txt. Raw MCC-Seq reads from the IUCPQ cohorts are deposited to the European Genome-phenome Archive (EGA) and available (accession no. EGAS00001003415) after approval by the Data Access Committee (DAC) designated to the study (https://www.ebi.ac.uk/ega/home). All other relevant data supporting the key findings of this study are available within the article and its Supplementary Information files or from the corresponding author upon reasonable request. A reporting summary for this Article is available as a Supplementary Information file.

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

## Acknowledgements

This work was supported by a Canadian Institute of Health Research (CIHR) team grant awarded to E.G. and M.L. (TEC-128093), a CIHR Foundation grant awarded to E.G (148391) and the CIHR funded Epigenome Mapping Center at McGill University (EP1-120608) awarded to T.P. and M.L. E.G. holds the Roberta D. Harding & William F. Bradley, Jr. Endowed Chair in Genomic Research and T.P. holds the Dee Lyons/Missouri Endowed Chair in Pediatric Genomic Medicine. A.T. is the director of a Research Chair in Bariatric and Metabolic Surgery. M.C.V. holds the Canada Research Chair in Genomics Applied to Nutrition and Health (Tier 1). F.A. held a studentship from The Fonds de recherche du Québec (FRSQ) during part of this study. The study was further supported by the Swedish Rheumatism Association and King Gustaf V's 80-years Foundation together with The Swedish Research Council and Wallenberg Foundation awarded to L.R. This study was also supported by the NIHR Oxford Biomedical Research Center. The views expressed are those of the authors and not necessarily those of the NHS, the NIHR or the Department of Health. The MuTHER Study was funded by a program grant from the Wellcome Trust (081917/Z/07/Z) and core funding for the Wellcome Trust Center for Human Genetics (090532). The TwinsUK study was funded by the Wellcome Trust and European Community's Seventh Framework Program (FP7/2007-2013). The TwinsUK study also receives support from the National Institute for Health Research (NIHR)—funded BioResource, Clinical Research Facility and Biomedical Research Center based at Guy's and St Thomas' NHS Foundation Trust in partnership with King's College London. We thank the NIH Roadmap Epigenomics Consortium and the Mapping Centers (http://nihroadmap.nih.gov/epigenomics/) for the production of publicly available reference epigenomes. Specifically, we thank the mapping centers at MGH/BROAD and UCSF for generation of human adipose (donor 92 and 7) and peripheral blood (TC014 and TC015) reference epigenomes used in this study, respectively. We further thank additional members of the MuTHER consortium for providing valuable data for this study. Please see the Supplementary Note in the Supplementary Information document for a full list of additional MuTHER members not already included in the author list.

## Author contributions

E.G. conceived the study. E.G., T.P., M.C.V., A.T., and M.L. designed experiments. F.G, L.R., A.T., and M.C.V. collected, prepared and/or provided the clinical samples. E.G. and F.A. lead data analyses. F.A., M.M.S., and E.B. performed experiments. F.A., A.K.H., W. A.C., X.S., and J.V. analyzed data. T.K. and B.G. provided bioinformatics support. A.K.

H., M.I.M., S.B., C.M., and P.D. provided replication data. F.A. generated figures with contributions from W.A.C. F.A. and E.G. drafted the manuscript. All authors reviewed and contributed feedback on the final manuscript.

## Additional information

**Competing interests:** A.T. receives research funding from Johnson & Johnson Medical Companies and Medtronic for studies unrelated to this manuscript. The remaining authors declare no competing interests.

