## [Peer Review File · Nature Communications]

Reviewers' comments:

Reviewer #1 (Remarks to the Author):

Allum and colleagues report on an EWAS of blood lipid levels in a sample of visceral fat biopsies from 199 morbidly obese individuals (bmi>40) who underwent bariatric surgery. Genome-wide DNA methylation was profiled using a capture bisulfite sequencing method (MCC-seq) resulting in 1.3M CpGs with sufficient coverage (33x) to evaluate. After discovery, the authors perform a broad range of analyses to explore lipid-associated CpGs. The small sample size, the fact that biopsies were taken from morbidly obese individuals, analytical issues and uncertainty about the causality of associations temper enthusiasm for the work.

- The authors claim that they measured 400 samples and 3M CpGs. The main analysis however is on 199 samples and 1.3M CpGs.

- Since biopsies were taken from morbidly obese individuals, visceral fat biopsies will most likely be highly variable in cell content between individuals for example due to infiltration of immune cells. The authors should provide histological data and adjust their analyses accordingly. More general: findings from this cohort cannot be generalized to the general population.

- The authors should adhere to current practise in GWAS. That is, control for cell counts, use proper methods to address (hidden) confounding and bias and inflation of test statistics (e.g. refer to Van Iterson. Genome Biol 2017). Also, they should provide QQ plots. The very high percentage of "significant" CpGs in their analysis (despite a small sample size) would be compatible with inflation and/or unaddressed confounding. Note that much larger scale lipid EWASes typically find dramatically fewer associated CpGs than reported here (over 2% of 1.3M CpGs tested).

- Validation of findings in Muthur should be presented right after discovery. Moreover, current thresholds are not convincing for 'replication' (same direction and nominal $P < 0.05$). For replication, a multiple testing correction should be applied. It will be good to note that Muthur sample are subcutaneous fat. They can do similar efforts using whole blood EWASes.

- In addition to these crucial analytical issues, the interpretation and biological significance of follow-up analyses is unclear (and the number of analyses is so large that it can be difficult to keep track of their purpose). Some examples:

- + Atypically, two FDR thresholds are used. In follow-up analyses, sometimes CpGs at FDR 0.05 and sometimes CpGs at FDR 0.01 are used. To prevent the suggestion that the authors used the threshold that led to the most convincing results, restrain to a single threshold. Preferably, use more conservative (and less sensitive to inflation) Bonferroni correction.

- + The authors noted that lipid associated CpGs are more variable than those who are not. Variability is a major determinant of power and will particularly in this small study drive the results. Since enhancers methylation in general is more variable than for example promoter methylation, enhancers and related annotations are bound to be enriched. The authors can use logistic regression for their enrichment analysis and include variance as a confounder (Bonder et al. Nat Genet 2017).

- + Published lipid showed that methylation changes are mostly induced by lipid levels. In this study, the direction for causality remains unclear.

- + A main claim of the authors claim is that MCC-seq is superior to methylation arrays. However, the method provides useful data for 1.3M CpGs which is not that much more than the 850k array and still only 4% of all CpGs in the human genome and probably at a much higher cost per sample. To gain insight in the putative value of MCC-seq, the annotation of the 1.3M should be provided and in addition can be compared with the EPIC and 450k arrays.

- + The authors discuss fine-mapping. It would be helpful to provide an analysis of the length of the regions (and no of CpGs) encompassed by EWAS signals in their analysis to provide insight in the utility of finemapping (instead of comparing with external 450k data). This may be more convincing than the current +/-250bp analyses comparing Muthur and discovery data.

- + Instead of inferring genotypes from sequencing data, it would be better to do direct genotyping or at least provide for a subset of samples an overview of the concordance rate with direct genotyping.

- + Tissue-independent CpGs were defined as multiple-testing significant in fat and nominally

significant in blood. If keeping this definition, then also include multiple-testing significant in blood and nominally significant in fat.

+ Adipose-specific: the absence of a nominal p-value <0.05 in blood cannot be interpreted as evidence that there is no difference in blood. Statistics cannot be used to show the absence of a difference.

Reviewer #2 (Remarks to the Author):

Allum et al. report a lipid-association study in a small sample of VAT with MCC-Seq data and subsequent replication in the MuTHER cohort (450k with low replication rate) and an additional blood sample with MCC-Seq data, followed by extensive enrichment testing. I liked the cross-tissue comparisons and also found the story around PPP2R1B interesting, both aspects are surely of relevance to the wider community. The manuscript is hard to read and rather difficult to grasp, particularly the motivations / aims for each section. A clear focus / major aim of the publication is missing. It could use a flow chart for the analyses clearly presenting those major aims, and perhaps a table indicating why each example presented shows the advantage of the MCC-seq method. I think in general the analyses are quite good, and most of the figures are as well, but it could be structured in a clearer manner. I found it difficult to move from the results section to the methods section. Similar, the introduction is quite lengthy while the discussion is not and some of the results are not picked up there.

1. The language in general is okay, but the sentences are sometimes long and complicated and there are more typos than would be acceptable, e.g. 48: typo: "sparse profiling ... has"; 74: typo: "heritability is"; 286: typo: "mapping to"; 700: typo: "into account" used twice; 702: typo: "except"; to mention only few.
2. The characteristics of epigenetic variants are well presented and the focus on LMRs and UMRs is interesting. The section on genetics could however be improved. What do we learn from the enrichment of SNP-CpGs among lipid-CpGs, given there seems to be no relationship of those SNPs to lipid levels? Is the greater density a unique feature of lipid CpGs? Given extended LD at some of these loci, why was the analysis restricted to 250kb (or 150kb later in the analysis)?
3. The positioning analysis is quite interesting, although not completely surprising. Other manuscripts previously reported differential methylation in promoter compared to gene body. Can this be replicated here?
4. The heading "Functional validation" is misleading and should be rephrased to "Functional annotation"
5. Line 154: confusing to use "up to 25%" but in the figure not present these percents, only fold changes
6. Line 157: "to putative enhancers active only in adipose tissue": how do they know they are only active there?
7. Line 238: "EPIC array captures 17% of CpGs profiled in LMRs by MCC-Seq compared to the 450K array,": The comparison (17%) is from EPIC to MCC-Seq, not to 450K, maybe change to "EPIC array captures 17% of CpGs profiled in LMRs using MCC-Seq; the 450K captures 6%"
8. Line 252: What does "delta methylation" mean? The range? Range should then be used again, for consistency.
9. Line 254: "Up to 23% of discovered lipid-CpGs were replicated in MuTHER at nominal p-value with same direction of effect (Methods) representing up to 1.3-fold-change (Fisher's $p < 3.7 \times 10^{-3}$)."
10. Line 268-270: Is it a test of proportions used?
11. Line 428/435/438: The authors seem to be using the word "GWAS" as a synonym for "GWAS SNP", something I have never seen before.
12. 480-492: Are these genes? Gene products? Shouldn't all gene names be italicized?

13. Several sentence fragments, e.g. sentence beginning "Thus, suggesting pleiotropic regulatory effects..." is a sentence fragment; or "Thus, hinting at a coordinated mechanistic regulation...".

14. Line 679: What does "a top 25th percentile cutoff in variance equating to +/-11.02% SD" mean?

15. Line 687: "Lipid levels not depicting a normal distribution were converted to the log scale": Which lipids were they?

Reviewer #1 (Anonymous):

General comments by the reviewer: Allum and colleagues report on an EWAS of plasma lipid levels in a sample of visceral adipose tissue biopsies from 199 morbidly obese individuals (bmi>40) who underwent bariatric surgery. Genome-wide DNA methylation was profiled using a capture bisulfite sequencing method (MCC-seq) resulting in 1.3M CpGs with sufficient coverage (33x) to evaluate. After discovery, the authors perform a broad range of analyses to explore lipid-associated CpGs. The small sample size, the fact that biopsies were taken from severely obese individuals, analytical issues and uncertainty about the causality of associations temper enthusiasm for the work

Response by the authors: We thank the reviewer for the careful evaluation and for valuable suggestions for the improvement of the paper. Although we concede that the sample size of the cohort presented in this study is smaller than some whole-blood based epigenomics studies in the literature, it does represent the first and largest cohort of the metabolic-disease linked visceral adipose tissue in terms of methylation profiles. While subcutaneous adipose tissues may more easily be obtained through skin biopsies and, thus, cohorts can be established in a population-based manner, obtaining visceral adipose tissue involves an invasive surgical intervention. Consequently, we are restricted to samples collected from obese individuals undergoing bariatric surgery, which also creates a limitation for acquiring large numbers of individuals. We thank the reviewer for their comment about the analytical issues and have now 1) conducted substantial re-analysis of all included data sets, 2) streamlined the manuscript to focus on the dissection of cardiometabolic-risk-linked adipose tissue regulatory regions and 3) included an additional cohort for improved insight into how identified cardiometabolic-linked epigenetic variants translate across disease states (i.e. obese vs. general population). We have further added a flow chart to permit readers to more easily tie-in the analyses towards this common goal (**Fig. 1**). Please see below details about the specific changes.

Specific comments by the reviewer:

1. The authors claim that they measured 400 samples and 3M CpGs. The main analysis however is on 199 samples and 1.3M CpGs.

Response by the authors: We apologize for grouping the sample numbers for different cohorts together and have now taken care to report the true numbers where appropriate (ex: p. 3, line 3). Similarly, we now only report the number of CpGs investigated in the discovery cohort (i.e. 1.3M; ex: p. 3, line 4).

2. Since biopsies were taken from morbidly obese individuals, visceral fat biopsies will most likely be highly variable in cell content between individuals for example due to infiltration of immune cells. The authors should provide histological data and adjust their analyses accordingly. More general: findings from this cohort cannot be generalized to the general population.

Response by the authors: We agree with the reviewer that EWAS of mixed tissues

samples (e.g. whole-blood, adipose tissue, brain) may pose challenges in interpreting results due to cellular heterogeneity. For whole-blood, multiple options are available to properly correct for individual cell composition including CBC count from lab tests or statistical deconvolution using inferred cell count in either a reference-based or reference-free setting. However, for complex tissues such as brain and adipose tissue no implementation has been presented to correct for cellular heterogeneity due to technological challenges in profiling the complete cellular landscape within these tissues. The suggestion by the reviewer to simply provide histological data and adjust accordingly is not feasible for two main reasons. First, adipose tissue collection either from population-based cohort or as used here from clinical cohorts are rare where only a handful of large-scale cohorts are to our knowledge available. Importantly, in both scenarios, tissue quantities are limited and together with the special expertise and resources needed for histological data, this is not done in a routine manner. In fact, we are unaware of any tissue collection of this size (of any complex tissue) where corresponding histological data is available. Secondly, in the unlikely scenario that such data would be available the procedure recommended by the reviewer is not straightforward. In fact, we have shown that there are substantial challenges in obtaining accurate cell distribution and measures of adipose tissue from histological data (PBMID: 26292076) and in comparison, with collagenase digestion (i.e. cell solution), the use of histological slides leads to lower values than expected. However, across all techniques tested we have shown that adipose tissue composition was associated with a general pattern of BMI-related increase with a plateau at higher obesity levels (i.e. those include in the current study). Taken together, this was the rationale for including BMI as a covariate in our model as a surrogate for adipose tissue composition.

Having said that, we nevertheless sought to address the reviewer's concern in further detail. Specifically, we spent substantial amount of time trying to develop a superior correction method for clinical obese cohorts and adipose tissue deconvolution than BMI-based corrections alone. First, we obtained matching visceral adipose tissue (VAT) samples

from the same cohort of individuals undergoing bariatric surgery and generated single-cell transcriptomes (10x Genomics) in an attempt to obtain a map of the cell types present. Then we used in-house and publicly available (<https://epigenomesportal.ca/ihec/>) NGS-based methylation data on VAT-specific cell types to select

Figure 1 QQplots of CpG to lipid (triglyceride) associations using cellular deconvolution (top, $\lambda=1.234$) versus BMI (bottom, $\lambda=1.038$). The complete test statistics was in both cases corrected for general inflation (BACON) as discussed below.

CpGs that were specifically hypomethylated (< 20% methylation in all samples for a cell type), which we then used to infer cell proportions in our VAT bulk MCC-Seq data from the ~200 subjects included in the study. Specifically, all EWAS of plasma lipid traits were re-analyzed including the inferred VAT cell counts in the GLM pipeline and compared with the original model that used BMI as a correction factor for cellular heterogeneity. Encouragingly, we found that the BMI-corrected model was superior as a correction factor with minimum inflation confirming our earlier work about the limited VAT variability among individuals at high BMI levels (ie >35kg/m²). Due to this, we present these former data to the reviewers only (see Figure 1 above) while focusing on BMI-corrected data in the presented paper.

3. The authors should adhere to current practice in GWAS. That is, control for cell counts, use proper methods to address (hidden) confounding and bias and inflation of test statistics (e.g. refer to Van Iterson. Genome Biol 2017). Also, they should provide QQ plots. The very high percentage of “significant” CpGs in their analysis (despite a small sample size) would be compatible with inflation and/or unaddressed confounding. Note that much larger scale lipid EWASes typically find dramatically fewer associated CpGs than reported here (over 2% of 1.3M CpGs tested).

Response by the authors: We again thank the reviewer for this valuable comment. We have now re-analyzed the complete data set and integrated the statistical method presented in Van Iterson *et al.*, which corrects for inflation of test statistics as suggested by the reviewer. We present the application of this method through QQplots before and after correction for each trait individually (**Supplementary Figures 1-4**). We observe that our QQplots now reflect a lack of inflation with all lambdas being very close to 1. We now focus exclusively on these more robust associations for downstream analyses. We note that all main conclusions previously reported before applying this method are maintained.

We believe that the additional increase in discovery power from this study may be attributable to the application of a targeted NGS approach in a tissue-relevant tissue. Most studies to date have applied array-based methods that are biased towards promoter regions whereas we have shown that trait-linked epigenetic variants are enriched in tissue-specific enhancers and promoters. Furthermore, past studies have used the bioavailable whole-blood tissue for discovery as opposed to more biologically-relevant tissues. These differences in study designs may have contributed to the observed differences in discovery power when contrasting past studies with our own study.

4. Validation of findings in Muther should be presented right after discovery. Moreover, current thresholds are not convincing for ‘replication’ (same direction and nominal P<0.05). For replication, a multiple testing correction should be applied. It will be good to note that Muther sample are subcutaneous fat. They can do similar efforts using whole blood EWASes.

Response by the authors: We thank the reviewer for pointing out this fact and have now moved up this section within the manuscript. However, we have kept an abbreviated

version of the section characterizing all lipid-linked regulatory regions before this MuTHER replication section. We believe that this section in part justifies the use of our method over other array-based methods. The MuTHER study is limited to 450K array covered loci with only 22% of adipose enhancer regions being sparsely covered, thereby, not permitting the same type of single-base analyses presented for all lipid-linked discovery regions. The MuTHER replication section is now modified and Bonferroni correction is applied for all traits individually (p. 11, line 3). We acknowledge that the MuTHER cohort adipose samples are subcutaneous fat whereas our discovery cohort samples are from the more disease-relevant visceral fat depot. As explained above, our cohort represents a new resource of epigenetic signals, which have not yet been examined. We use the MuTHER cohort here as the best alternative proxy large-scale resource of available data and are limited to pursue epigenetic variants stable across these two tissue depots. We believe that using VAT as the starting material points to more biologically-relevant loci after replication. In addition, we have added a second replication cohort in whole-blood (CARTaGENE) profiled using the same platform (MCC-Seq) and applying a multiple correction approach as suggested by the reviewer that more closely mirrors epigenetic changes in the general population (p. 15, line 16).

5. In addition to these crucial analytical issues, the interpretation and biological significance of follow-up analyses is unclear (and the number of analyses is so large that it can be difficult to keep track of their purpose). Some examples:

+ Atypically, two FDR thresholds are used. In follow-up analyses, sometimes CpGs at FDR 0.05 and sometimes CpGs at FDR 0.01 are used. To prevent the suggestion that the authors used the threshold that led to the most convincing results, restrain to a single threshold. Preferably, use more conservative (and less sensitive to inflation) Bonferroni correction.

Response by the authors: We thank the reviewer for their comment. We have now applied the stringent *BACON* statistical method to correct for inflation of our EWAS distribution as discussed above. We additionally tabulated q-values of the EWAS per lipid trait and now utilize a fixed cutoff of FDR that is consistent throughout the study. We believe that layering multiple quantitative analyses at this threshold through three-way associations of methylation, expression and lipids provides the additional confidence needed for the main conclusions. Furthermore, for all replications in the MuTHER and blood cohorts, Bonferroni correction is now applied in order to stay within the norms of published works as recommended by the reviewer.

+ The authors noted that lipid associated CpGs are more variable than those who are not. Variability is a major determinant of power and will particularly in this small study drive the results. Since enhancers methylation in general is more variable than for example promoter methylation, enhancers and related annotations are bound to be enriched. The authors can use logistic regression for their enrichment analysis and include variance as a confounder (Bonder et al. Nat Genet 2017).

Response by the authors: We are very much in agreement with the reviewer and

acknowledge that variability may play a role in the positional enrichment findings. As such, in order to counter for this, we do additionally investigate these main conclusions for the top variable CpG set (i.e. top 25% variable CpGs in terms of methylation status across all 199 individuals; p. 7, line 13). As exemplified in **Supplementary Figure 7**, we show that enrichments in enhancers and tissue-specific elements and depletion in promoters is not only maintained but strengthened in the variable set.

+ Published lipid showed that methylation changes are mostly induced by lipid levels. In this study, the direction for causality remains unclear.

Response by the authors: We acknowledge this current trend in methylation-related publications and do not dispute these findings. However, a main difference with our manuscript compared to those that the reviewer refers to is the additional focus on epigenetic variations under non-genetic. Importantly, strong associations between SNPs and DNA methylation are needed to make good instruments in analysis such as Mendelian Randomization (MR) in order to distinguish between spurious associations resulting from confounding factors and a potentially causal relationship. To this end, we show that a significant proportion (i.e. 51%; p. 18, line 16) of adipose-specific epigenetic variation (i.e. not shared across tissues to whole-blood) has no or very minor genetic associations and, thus, MR models are not applicable. Instead, we apply a three-way association approach of methylation, expression and lipid requiring the replicated lipid-CpG to be associated with gene expression which in turn associates with the same lipid trait under investigation. We follow-up on an example on chromosome 19 (p. 18, line 18) where we identify an adipose-specific enhancer harboring CpGs whose methylation levels is significantly associated with lipid levels across discovery and replication cohort (e.g. $p=5.1E-10$ in the large-scale MuTHER cohort). Through our comprehensive three-way association approach as described above we show that the enhancer is regulating the expression of multiple genes with the strongest links exhibited to *GNAI5* and *GNG7* expression. Independent associations of the expression of these two genes in the large MuTHER adipose cohort with lipid levels confirmed the link between epigenetic regulation of the locus with levels of TG and the disease-state. Finally, we link the identified co-regulation network to the $G\alpha_q$ Signaling pathway, which was enriched among our discovered epigenetically regulated genes.

+ A main claim of the authors claim is that MCC-seq is superior to methylation arrays. However, the method provides useful data for 1.3M CpGs which is not that much more than the 850k array and still only 4% of all CpGs in the human genome and probably at a much higher cost per sample. To gain insight in the putative value of MCC-seq, the annotation of the 1.3M should be provided and in addition can be compared with the EPIC and 450k arrays.

Response by the authors: We agree with the reviewer that the difference in absolute quantities of CpGs between our restricted MCC-Seq panel used here (i.e. ~1.3M CpGs) and the Illumina EPIC (i.e. ~850K CpGs) may appear insignificant. However, the

important difference lies within the regions covered by the methods, where the EPIC array is designed as a “one-size-fits-all” approach. In fact, we found that even for the most commonly used tissue type (i.e. blood), only 54% of enhancers that are active in this tissue are covered by at least one CpG indicating that a large and important part of the functional methylome in blood is not included on the EPIC array. In addition, although the EPIC targets up to half of the active enhancers in whole-blood, we note that the density coverage within these regulatory elements is quite scarce. We also show in the presented paper, that only 17% of the total percent CpGs profiled with MCC-Seq at adipose enhancers are included on the EPIC array (p. 10, line 9). We further highlight clear ascertainment biases of this method within adipose regulatory elements – especially at promoter regions (p. 10, line 13; **Supplementary Figure 9**). Finally, the 1.3M CpGs presented throughout the paper (of the total 3.3M profiled) represent the most highly covered and variable CpGs within this cohort, indicating the significant added value of in-depth MCC-Seq profiling at regulatory regions in comparison to the EPIC array. We have now included a table (**Supplementary Table 4**) of the 1.3M CpGs indicating which CpGs are also directly typed on the EPIC and MuTHER arrays.

+ The authors discuss fine-mapping. It would be helpful to provide an analysis of the length of the regions (and no of CpGs) encompassed by EWAS signals in their analysis to provide insight in the utility of finemapping (instead of comparing with external 450k data). This may be more convincing than the current +/-250bp analyses comparing Muther and discovery data.

Response by the authors: We thank the reviewer for this suggestion. Length analyses of the different types of regions (i.e. LMRs and UMRs) were published by our group in a recent publication (*Busche et al.*, 2015). We have re-summarized these results in the presented manuscript along with the genomic density of discovery CpGs at these regions (**Supplementary Table 3**).

+ Instead of inferring genotypes from sequencing data, it would be better to do direct genotyping or at least provide for a subset of samples an overview of the concordance rate with direct genotyping.

Response by the authors: We previously presented the accuracy and specificity of MCC-Seq to simultaneously provide genotype calls over target regions in a previous publication in *Nature Communications* (*Allum et al.*, 2015). We refer to this publication in the manuscript where appropriate (p. 4, line 21; p. 16, line 6) in order to point the reader towards the proper information. We have appended an excerpt from our 2015 article that supports on this validation for the reviewer;

“The same 24 AT samples described above were also genotyped with the Illumina HumanOmni2.5S-8 BeadChip array for validation of MCC-Seq's ability to simultaneously call genotypes. After stringent quality control, we obtained SNP genotypes at 94,600 overlapping loci using MCC-Seq (Met V1) (Methods). We observed 99% genotype concordance between the two methods at sites on the SNP array, indicating that MCC-Seq has the potential

to allow for simultaneous and accurate genotyping calling over regions of interest. Similarly, comparing the observed heterozygosity from the two measurements yielded high correlation”

+ Tissue-independent CpGs were defined as multiple-testing significant in fat and nominally significant in blood. If keeping this definition, then also include multiple-testing significant in blood and nominally significant in fat.

Response by the authors: We have now streamlined the analyses in the manuscript and have dropped this additional concept of “tissue-independent” CpGs. We now focus on lipid-linked regulatory regions replicating in our whole-blood cohorts for dissection of the tissue-specific epigenetic signature observed (p. 14, line 1).

+ Adipose-specific: the absence of a nominal p-value <0.05 in blood cannot be interpreted as evidence that there is no difference in blood. Statistics cannot be used to show the absence of a difference.

Response by the authors: Please see above, we have now streamlined the analyses in the manuscript and have dropped this additional concept of “tissue-specific CpGs”.

Reviewer #2 (Anonymous):

General comments by the reviewer: Allum et al. report a lipid-association study in a small sample of VAT with MCC-Seq data and subsequent replication in the MuTHER cohort (450k with low replication rate) and an additional blood sample with MCC-Seq data, followed by extensive enrichment testing. I liked the cross-tissue comparisons and also found the story around PPP2R1B interesting, both aspects are surely of relevance to the wider community. The manuscript is hard to read and rather difficult to grasp, particularly the motivations / aims for each section. A clear focus / major aim of the publication is missing. It could use a flow chart for the analyses clearly presenting those major aims, and perhaps a table indicating why each example presented shows the advantage of the MCC-seq method. I think in general the analyses are quite good, and most of the figures are as well, but it could be structured in a clearer manner. I found it difficult to move from the results section to the methods section. Similar, the introduction is quite lengthy while the discussion is not and some of the results are not picked up there.

Response by the authors: We thank the reviewer for their careful evaluation and for valuable suggestions for the improvement of the paper. We have addressed the concerns raised by the reviewer by streamlining the flow of the paper. We have additionally added a flow chart to permit readers to more easily tie-in the analyses towards this common goal (**Fig. 1**). We have now further highlight the advantages of MCC-Seq throughout the manuscript and most prominently in the discussion section. Please see below details about the specific changes implemented in the paper.

Specific comments by the reviewer:

1. The language in general is okay, but the sentences are sometimes long and complicated and there are more typos than would be acceptable, e.g. 48: typo: “sparse profiling ... has”; 74: typo: “heritability is”; 286: typo: “mapping to”; 700: typo: “into account” used twice; 702: typo: “except”; to mention only few.

Response by the authors: We thank the reviewer for their comment and have now taken the time to review the grammar and syntax in the manuscript carefully.

2. The characteristics of epigenetic variants are well presented and the focus on LMRs and UMRs is interesting. The section on genetics could however be improved. What do we learn from the enrichment of SNP-CpGs among lipid-CpGs, given there seems to be no relationship of those SNPs to lipid levels? Is the greater density a unique feature of lipid CpGs? Given extended LD at some of these loci, why was the analysis restricted to 250kb (or 150kb later in the analysis)?

Response by the authors: We have now added a paragraph within the section entitled “**Genetic contribution to lipid-CpG methylation variability**”, which overlaps GWAS SNPs for the same plasma lipid traits under study from the large-scale efforts of the Global Lipids Consortium (Willer *et al*, 2013) with top SNPs regulating methylation at our lipid-linked regulatory regions. Briefly, we show that there is an enrichment of the SNPs from this fully released dataset among our metQTL SNPs at different threshold of significance (p. 18, line 1).

We have now streamlined the analyses presented in this manuscript and, consequently, have opted to omit the genomic density analyses performed for genetic versus non-genetic lipid-CpGs.

We have opted to limit to a range of +/-250kb for our SNP-CpG associations for the following reasons; (1) as shown in **Supplementary Figure 13** there is an enrichment of SNPs associated with CpG methylation within the vicinity of their linked CpGs; (2) this genomic window is in line with multiple other studies; (3) we were keen to utilize this genomic set previously released from our group and were limited to this genomic window.

3. The positioning analysis is quite interesting, although not completely surprising. Other manuscripts previously reported differential methylation in promoter compared to gene body. Can this be replicated here?

Response by the authors: We thank the reviewer for their interest in these analyses. Most publications previously released have utilized array-based methods to investigate differential methylation status across the genome. We show in **Supplementary Figure 9** that these methods depict a clear lack of coverage at the gene body region near the transcription start site (TSS). This may explain the observation put forth by the reviewer.

In our manuscript, we are emphasizing that differential methylation linked to quantitative traits seems to locate to both upstream and downstream (gene body region) of the TSS (**Figure 1B**) with a slight shift towards the latter region. When taking the whole promoter region versus gene body region as a whole (i.e. instead of the regions directly flanking the TSS), it was previously reported by our group that the gene body region actually harbors more variable CpGs than the promoter region (Grundberg *et al.*, 2013).

4. The heading “Functional validation” is misleading and should be rephrased to “Functional annotation”

Response by the authors: The title of this section is now amended to "**Functional annotation of lipid-CpGs**" (p. 12, line 3).

5. Line 154: confusing to use “up to 25%” but in the figure not present these percents, only fold changes

Response by the authors: We have maintained this format in reporting as we believe that the reader can benefit from reading both a comparison in terms of percentages with the background as well as a bar graph representation of the fold-change to better contrast between the observed changes at the different types of regulatory elements.

6. Line 157: “to putative enhancers active only in adipose tissue”: how do they know they are only active there?

Response by the authors: We have now reworded this sentence to “adipose regulatory region not shared with other tissues (i.e. whole-blood)” (p. 7, line 7) and now refer to this set as “adipose-specific” subsequently. We further now refer to the Methods section for an explanation of how this set was previously defined (Allum *et al.*, 2015; Busche *et al.*, 2015).

7. Line 238: “EPIC array captures 17% of CpGs profiled in LMRs by MCC-Seq compared to the 450K array,”: The comparison (17%) is from EPIC to MCC-Seq, not to 450K, maybe change to “EPIC array captures 17% of CpGs profiled in LMRs using MCC-Seq; the 450K captures 6%”

Response by the authors: We have now reworded this sentence to; “As a whole, we tabulated that the EPIC array and 450K array captured only 17% and 6% of the total percent CpGs profiled in LMRs by MCC-Seq and 29% and 19% of those mapping to UMRs, respectively.” (p. 10, line 9).

8. Line 252: What does “delta methylation” mean? The range? Range should then be used again, for consistency.

Response by the authors: We are now using the term "range" instead of "delta methylation" throughout the manuscript (ex: **Supplementary Figure 6**).

9. Line 254: “Up to 23% of discovered lipid-CpGs were replicated in MuTHER at nominal p-value with same direction of effect (Methods) representing up to 1.3-fold-change (Fisher’s $p < 3.7 \times 10^{-3}$).” Where are the results?

Response by the authors: We apologize for this omission. The annotated replicated discovery lipid-linked regions are now presented in **Supplementary Table 5**. We have also increased the stringency of replication to fit with the norms of the field.

10. Line 268-270: Is it a test of proportions used?

Response by the authors: Yes, the Binomial test is used for this analysis (p. 14, line 13).

11. Line 428/435/438: The authors seem to be using the word “GWAS” as a synonym for “GWAS SNP”, something I have never seen before.

Response by the authors: We thank the reviewer for pointing out this mistake. We are now using the term "GWAS SNP" instead of "GWAS" were appropriate.

12. 480-492: Are these genes? Gene products? Shouldn’t all gene names be italicized

Response by the authors: The manuscript is now amended so that all gene names are italicized.

13. Several sentence fragments, e.g. sentence beginning “Thus, suggesting pleiotropic regulatory effects...” is a sentence fragment; or “Thus, hinting at a coordinated mechanistic regulation...”.

Response by the authors: We apologize for these mistakes in syntax and have now corrected these types of sentences.

14. Line 679: What does “a top 25th percentile cutoff in variance equating to +/-11.02% SD” mean?

Response by the authors: We have streamlined the analyses presented in the manuscript and now omit the section that included this statement.

15. Line 687: “Lipid levels not depicting a normal distribution were converted to the log scale”: Which lipids were they?

Response by the authors: We apologize for this omission in detail and have now updated the Methods section to specify which lipid traits were converted to the log scale within each cohort (p. 35, line 1).

REVIEWERS' COMMENTS:

Reviewer #1 (Remarks to the Author):

Allum and colleagues put a significant effort in revising their manuscript. I appreciate the new version and I enjoyed reading about the findings. While reading several things came up:

- Using an FDR threshold of 10% as the authors do in for their discovery analysis is uncommon. I understand that they explore this larger set and continue the analysis in more detail only after replication, in which case the original p-value has become irrelevant. I think the authors should note the discussion that their FDR threshold is more liberal than usual and elaborate why they believe this is warranted in their study. Moreover, the authors should in suppl figures 1-4 colour all CpGs FDR<5% and 10%<FDR<5%. This will allow the readers to make their own judgement. Were the enrichments for putative adipose enhancers different for FDR<5% than FDR<10%?
- The authors now included sequencing data in whole blood. When comparing VAT and whole blood, the authors talk about replication and validation. While non-replication in general is a bad thing, here this will not be the case. More neutral terms like 'evidence for tissue-shared association' may be applicable.
- With respect to the latter: 'replication' between VAT and blood indicates that the differential methylation is the consequence rather than the cause. After all, blood cells are no source of circulating lipids. This can be noted. Similarly, adipose tissue is an important source of circulating TGs. Is this also the case for HDL and LDL?
- Finally, the authors find interesting overlaps between regions in which methylation is associated with lipids and lipid-associated SNPs discovered in GWAS. However, it can then not be ruled out that the methylation difference is driven by the GWAS SNP. Did the differential methylation remain after correction for the SNP the authors found to be associated with DNA methylation in their study?'

Reviewer #2 (Remarks to the Author):

The point by point response letter satisfactorily adressed my previous comments.

REVIEWERS' COMMENTS:

Reviewer #1 (Remarks to the Author):

Allum and colleagues put a significant effort in revising their manuscript. I appreciate the new version and I enjoyed reading about the findings. While reading several things came up:

1. Using an FDR threshold of 10% as the authors do in for their discovery analysis is uncommon. I understand that they explore this larger set and continue the analysis in more detail only after replication, in which case the original p-value has become irrelevant. I think the authors should note the discussion that their FDR threshold is more liberal than usual and elaborate why they believe this is warranted in their study. Moreover, the authors should in suppl figures 1-4 colour all CpGs $FDR < 5\%$ and $10\% < FDR < 5\%$. This will allow the readers to make their own judgement. Were the enrichments for putative adipose enhancers different for $FDR < 5\%$ than $FDR < 10\%$?

Response by the authors: We thank the reviewer for their comments. We have now added vertical lines in Supplementary Figures 1-4 to indicate $FDR < 5\%$ and $FDR < 10\%$ cutoffs. We have additionally amended the discussion to include the following statement (p.18): “We combine a stringent statistical correction method (BACON) with a more lenient FDR threshold (10%) and perform detailed follow-ups on regions replicating across adipose tissue depots and across tissue types (whole blood) using the classical Bonferroni approach. This strategy allowed us to present an expanded resource of cardiometabolic risk-linked epigenetic loci.”

Enrichment trends noted at $FDR < 10\%$ (1.5-fold-change; Fisher's $p = 6.6E-13$) in putative adipose enhancer regions were maintained at $FDR < 5\%$ (1.4-fold-change; Fisher's $p = 6.2E-05$). We believe that this slight drop in fold-enrichment may be attributable to the smaller number of sites being investigated at $FDR < 5\%$ ($N = 615$) versus at $FDR < 10\%$ ($N = 1,230$).

2. The authors now included sequencing data in whole blood. When comparing VAT and whole blood, the authors talk about replication and validation. While non-replication in general is a bad thing, here this will not be the case. More neutral terms like ‘evidence for tissue-shared association’ may be applicable.

Response by the authors: We have now modified the following sentence (p.12) “Taken together, we identified 68 adipose tissue regulatory regions (13 putative enhancers and 55 promoters) harboring lipid-associations that replicated in whole-blood.” to “Taken together, we identified 68 adipose tissue regulatory regions (13 putative enhancers and 55 promoters) showing evidence for tissue-shared lipid-associations.”

3. With respect to the latter: ‘replication’ between VAT and blood indicates that the differential methylation is the consequence rather than the cause. After all, blood cells are no source of circulating lipids. This can be noted. Similarly, adipose tissue is an important source of circulating TGs. Is this also the case for HDL and LDL?

Response by the authors: We thank the reviewer for their comment. We would like to emphasize that the lipid-linked regions that show evidence for tissue-shared associations are enriched in putative genetic effects over non-shared lipid-linked regions. This finding indicates that these observed lipid-associations may be independent of local lipid effects and, instead, have a basis in mechanisms orchestrated by an individual's genetic background in response to global changes in circulating lipid levels. In this way, we present a subset of lipid-linked regulatory regions detectable across adipose to a bioavailable tissue, which may be more easily assessed in a clinical setting.

4. Finally, the authors find interesting overlaps between regions in which methylation is associated with lipids and lipid-associated SNPs discovered in GWAS. However, it can then not be ruled out that the methylation difference is driven by the GWAS SNP. Did the differential methylation remain after correction for the SNP the authors found to be associated with DNA methylation in their study?

Response by the authors: We agree with the reviewer that in these instances the GWAS SNP may be driving the methylation status and, thus, currently refer to these regions as being under genetic regulation. As suggested by the reviewer, we have now added a conditional analyses model for the SNPs showing putative regulatory effects on the adipose-specific enhancer on *GALNT2*; “Genetic effects at this enhancer were supported by conditioning on these 21 SNPs, independently, where reduction of lipid-association significance was noted for all SNPs with rs2760537 being the most prominent (corrected $p=4.3 \times 10^{-2}$; $q=0.78$; Methods).” (p.18).

Reviewer #2 (Remarks to the Author):

The point by point response letter satisfactorily addressed my previous comments.